# INFONCE INDUCES GAUSSIAN DISTRIBUTION

**Roy Betser**[*]   **Eyal Gofer**[*]   **Meir Yossef Levi**   **Guy Gilboa**
**Technion - Israel Institute of Technology**

## ABSTRACT

Contrastive learning has become a cornerstone of modern representation learning, allowing training with massive unlabeled data for both task-specific and general (foundation) models. A prototypical loss in contrastive training is InfoNCE and its variants. In this work, we show that the InfoNCE objective induces Gaussian structure in representations that emerge from contrastive training. We establish this result in two complementary regimes. First, we show that under certain alignment and concentration assumptions, projections of the high-dimensional representation asymptotically approach a multivariate Gaussian distribution. Next, under less strict assumptions, we show that adding a small asymptotically vanishing regularization term that promotes low feature norm and high feature entropy leads to similar asymptotic results. We support our analysis with experiments on synthetic and CIFAR-10 datasets across multiple encoder architectures and sizes, demonstrating consistent Gaussian behavior. This perspective provides a principled explanation for commonly observed Gaussianity in contrastive representations. The resulting Gaussian model enables principled analytical treatment of learned representations and is expected to support a wide range of applications in contrastive learning.

## 1   INTRODUCTION

Self-supervised learning with contrastive objectives has transformed modern representation learning, enabling scalable training of encoders without labels (Oord et al., 2018; Chen et al., 2020; He et al., 2020; Radford et al., 2021). Among these objectives, the InfoNCE loss balances two pressures: positive pairs are aligned while the batch is repelled to encourage uniformity (Wang & Isola, 2020). This uniformity is often described geometrically as "spreading out" the data on the hypersphere (Chen & He, 2021), but a deeper probabilistic question remains: *What is the actual distribution of representations trained with InfoNCE?*

Answering this question is not only of theoretical interest. A Gaussian characterization is directly motivated by recent empirical findings suggesting that "more Gaussian" representations can correlate with improved downstream performance (Eftekhari & Papyan, 2025). It also provides a principled basis for practical methods that model contrastive representations as Gaussians for tasks such as classification, uncertainty estimation and test-time adaptation (Baumann et al., 2024; Morales-Álvarez et al., 2024). Moreover, assuming Gaussian structure makes quantities such as entropy, likelihood and KL divergences available in closed form, which underpins density-based diagnostics (Lee et al., 2018; Betser et al., 2025). These benefits are already exploited in applied work, with recent studies empirically observing and leveraging approximate Gaussian behavior in self-supervised representations (Baumann et al., 2024; Balestriero et al., 2025; Betser et al., 2026). Yet, despite these developments, a principled population-level explanation of why contrastive objectives such as InfoNCE give rise to Gaussian structure in representation space remains lacking.

Analyzing the *population* InfoNCE objective, we formalize the emergence of asymptotically Gaussian representations through two complementary analytical routes. A key ingredient is a novel alignment bound based on Hirschfeld-Gebelein-Rényi (HGR) maximal correlation, which limits achievable alignment according to augmentation mildness (Sec. 3.1). In the *empirical idealization route*, motivated by empirical training dynamics, alignment reaches a plateau and the objective reduces to a constrained uniformity problem on the hypersphere; combined with norm concentration, this yields Gaussian structure for both normalized (to a unit norm) and unnormalized representations (Sec. 4.1).

---

[*]These authors contributed equally to this work.
Corresponding author: `roybe@campus.technion.ac.il`

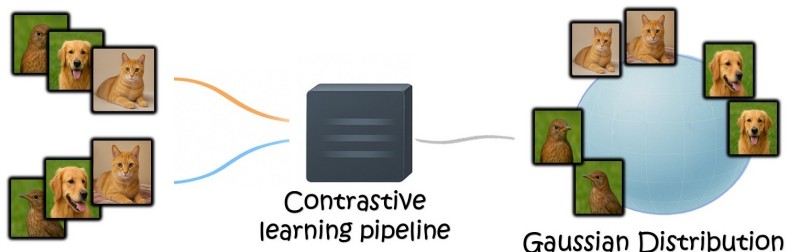

Figure 1: **Illustration.** Contrastive learning yields (approximately) Gaussian representations.

In the *regularized route*, a population-level analysis shows that adding a vanishing convex regularizer prioritizes the isotropic solution, yielding the same asymptotic Gaussian behavior without relying on training dynamics (Sec. 4.2). Together, these analyses shed light on why Gaussian structure can emerge under the InfoNCE objective at the population level.

We complement our theoretical analysis with empirical studies on synthetic data and CIFAR-10 (Krizhevsky et al., 2009) images, using encoders of increasing complexity: linear layers, MLPs with nonlinear activations, and ResNet-18 (He et al., 2016). By comparing contrastive and supervised training, we isolate the role of the training objective, beyond effects of data or architecture. We further observe similar Gaussian statistics in representations learned by general self-supervised foundation models, including DINO (Caron et al., 2021), motivating a broader examination of Gaussian structure across self-supervised objectives. Our **main contributions** are:

- **Bounded alignment.** In the large-batch limit, the alignment induced by the InfoNCE objective is bounded by the strength of the data augmentations.
- **Uniformity on the sphere.** Along both routes we analyze, normalized representations converge toward the uniform distribution on the unit sphere.
- **Asymptotic Gaussian structure.** Within this framework, both normalized and unnormalized representations admit asymptotically Gaussian behavior under the InfoNCE objective.
- **Empirical support.** Accompanying our asymptotic analysis, we provide finite-dimensional empirical evidence on synthetic and real data, illustrating the emergence of Gaussian behavior across multiple settings and encoder architectures.

## 2 RELATED WORK

**Contrastive learning and InfoNCE.** The InfoNCE loss (Oord et al., 2018) is the standard objective in self-supervised representation learning and underlies methods such as SimCLR (Chen et al., 2020), MoCo (He et al., 2020), and CLIP (Radford et al., 2021). It balances alignment of positive pairs with batch-wise repulsion that promotes uniformity in representation space (Wang & Isola, 2020; Chen & He, 2021). Prior work has studied these effects from geometric and optimization perspectives, identifying phenomena such as hyperspherical uniformity and feature concentration (Chen & He, 2021; Caron et al., 2021; Draganov et al., 2025). Other empirical studies model contrastive representations as approximately Gaussian (Baumann et al., 2024; Morales-Álvarez et al., 2024). However, the probabilistic law induced by the InfoNCE objective itself remains theoretically unexplained.

**Isotropy and Gaussian structure.** Several works aim to promote isotropic or Gaussian-like representations through explicit regularization or architectural design, including whitening-based objectives, variance-covariance control, and neural collapse phenomena (Ermolov et al., 2021; Papyan et al., 2020; Bardes et al., 2022). Related self-supervised approaches based on joint-embedding predictive architectures (JEPA) also yield highly regular representations and have been shown to encode density-related structure that can be exploited with Gaussian models (Assran et al., 2023; Bardes et al., 2024; Balestriero et al., 2025; Balestriero & LeCun, 2025). However, these works primarily observe or exploit Gaussian-like structure rather than explain its origin. Our work instead shows how Gaussianity emerges directly from the population InfoNCE objective.

**Hyperspherical geometry and Gaussianity.** A classical body of work studies the geometry of high-dimensional uniform measures on the sphere and their connection to Gaussian distributions (Vershynin, 2018; Wegner, 2021). Related geometric ideas also appear in hyperspherical variational families and radial Bayesian priors, which leverage approximately uniform distributions over the hypersphere (Davidson et al., 2018; Farquhar et al., 2020). A central result in this literature is the Maxwell-Poincaré spherical central limit theorem, which shows that fixed-dimensional projections of the uniform distribution on $\mathbb{S}^{d-1}$ converge to a Gaussian as the dimension grows (Maxwell, 1860; Poincaré, 1912; Diaconis & Freedman, 1987). Although developed independently of contrastive learning, these results provide the mathematical basis for why spherical uniformity induces Gaussian structure in high-dimensional representations. Our analysis connects this classical theory to contrastive learning by identifying regimes in which the InfoNCE objective induces such uniformity.

**Additional theoretical perspectives.** Complementary lines of work study theoretical properties of representations learned with contrastive objectives. Identifiability analyses characterize when latent variables or semantic factors can be uniquely recovered under structural assumptions on the data-generating process (Hyvarinen & Morioka, 2016; Hyvarinen et al., 2019; Zimmermann et al., 2021; Roeder et al., 2021; Reizinger et al., 2024); these results concern conditional or component-level structure and do not make claims about the marginal distribution of representations. Separately, task-driven analyses establish class separability or clustering guarantees for contrastive representations (Saunshi et al., 2019; HaoChen et al., 2021), focusing on class-conditional geometry rather than the overall distribution. Concretely, class-specific clusters may remain well separated even when the overall embedding distribution is approximately Gaussian. Our work does not address recovery or class structure; instead, it analyzes the marginal distribution induced by the population InfoNCE objective.

## 3 SETUP

**Data domain.** Let $(\mathcal{X}, \mathcal{B}(\mathcal{X}))$ be a standard Borel space (a standard setting in probability) with a base probability $p_{\text{base}}$. We draw $X_0 \sim p_{\text{base}}$ as a single data item (e.g., an image).

**Pairs via augmentation.** Contrastive learning is built around pairs of related examples rather than individual samples. To form such pairs, we use an *augmentation channel* $\mathcal{A}$, which takes a base sample $X_0 \sim p_{\text{base}}$ and produces stochastic variations of it. Formally, given $X_0$, we draw two independent augmentations

$$X, Y \sim \mathcal{A}(\cdot \mid X_0). \tag{1}$$

Here $X$ and $Y$ are two views of the same underlying example (e.g., different crops or color jitter). We denote by $p_X$ the marginal distribution of a single augmentation and assume it is nonatomic (a mild technical condition achievable in practice by infinitesimal dither). $p_{XY}$ denotes the joint distribution of a pair of augmentations $(X, Y)$.

**InfoNCE loss.** Let $f : \mathcal{X} \to \mathbb{R}^d$, $d \geq 2$, be a Borel-measurable encoder that maps input data to representations. InfoNCE operates on $\ell_2$-normalized representations, defined as $\hat{f}(x) := f(x)/\|f(x)\|$ if $\|f(x)\| > 0$, and $\hat{f}(x) := c_0$ for a fixed arbitrary $c_0 \in \mathbb{S}^{d-1}$ otherwise. Given a batch of $N$ paired augmentations $\{(x_i, y_i)\}_{i=1}^N$ drawn i.i.d. from $p_{XY}$, define $u_i := \hat{f}(x_i)$ and $v_i := \hat{f}(y_i)$. The empirical InfoNCE loss is

$$\mathcal{L}_{\text{InfoNCE}} = -\frac{1}{N} \sum_{i=1}^N \log \frac{\exp\left(\frac{1}{\tau}\langle u_i, v_i \rangle\right)}{\sum_{j=1}^N \exp\left(\frac{1}{\tau}\langle u_i, v_j \rangle\right)}, \tag{2}$$

with a fixed temperature $\tau > 0$. Since $u_i$ and $v_j$ are unit-normalized, $\langle u_i, v_j \rangle$ equals cosine similarity. The numerator measures the similarity of the *positive* pair $(u_i, v_i)$. The denominator compares each anchor $u_i$ to all candidates $\{v_j\}_{j=1}^N$, where $j \neq i$ serve as *negatives*. This softmax encourages $u_i$ to rank its true partner highest while remaining distinct from negatives, preventing collapse.

**Population InfoNCE.** The empirical InfoNCE loss in Eq. (2) depends on the batch size $N$. As $N \to \infty$, the empirical averages converge to expectations. Let

$$\mu := \hat{f}_* p_X, \qquad \pi := (\hat{f}, \hat{f})_* p_{XY}, \tag{3}$$

be the marginal distribution of representations and the joint distribution of positive pairs, respectively. Here $\hat{f}_* p_X$ denotes the *pushforward measure* of $p_X$ by $\hat{f}$, which is the distribution of $\hat{f}(X)$. As shown by Wang & Isola (2020, Theorem 1, Eq. (2)), in the infinite-negatives limit $N \to \infty$ the empirical InfoNCE loss (up to the additive $\log N$ term) converges to the following population functional. With $\alpha = 1/\tau$ for fixed $\tau > 0$:

$$\mathcal{L}(\mu, \pi) \;=\; -\alpha \, \mathbb{E}_{(u,v) \sim \pi}[u \cdot v] \;+\; \Phi(\mu), \qquad \Phi(\mu) := \mathbb{E}_{u \sim \mu} \log \mathbb{E}_{v \sim \mu} \exp\!\big(\alpha \, u \cdot v\big). \tag{4}$$

The first term measures *alignment* of positive pairs, while the second is a *uniformity potential* depending only on $\mu$.

## 3.1 ALIGNMENT BOUND

We now introduce a new term that quantifies the degree of augmentation. The augmentation channel $\mathcal{A}$ limits how much positive-pair alignment can be induced. We quantify this with the *augmentation mildness* parameter

$$\eta_2 \;:=\; \sup_{\substack{g \in L^2(p_X) \\ \mathrm{Var}(g) > 0}} \frac{\mathrm{Var}\big(\mathbb{E}[g(X) \mid X_0]\big)}{\mathrm{Var}(g(X))} \;\in [0, 1], \tag{5}$$

which measures how predictable functions of the view $X$ are from the base $X_0$. This quantity equals the squared Hirschfeld-Gebelein-Rényi (HGR) maximal correlation, denoted $\rho_m(X, X_0)$, i.e., $\eta_2 = \rho_m^2(X, X_0)$ (Hirschfeld, 1935; Gebelein, 1941; Rényi, 1959) (see Appendix A.1). Intuitively, $\eta_2 = 0$ when $X$ is (effectively) independent of $X_0$ (very strong/noisy augmentations), and $\eta_2 = 1$ when $X$ is fully determined by $X_0$ (no augmentation noise).

**Example.** Consider the Gaussian channel $X = A X_0 + \sqrt{1 - A^2}\, \varepsilon$, where $X_0 \sim \mathcal{N}(0, 1)$ and $\varepsilon \sim \mathcal{N}(0, 1)$ are independent. In this case, $X$ and $X_0$ are jointly Gaussian with Pearson correlation $A$, the maximal correlation satisfies $\rho_m(X, X_0) = |A|$, and thus $\eta_2 = A^2$ (Appendix A.2).

**Proposition 1** (Augmentation-controlled alignment bound)**.** *Let $X, Y \sim \mathcal{A}(\cdot \mid X_0)$ be conditionally independent given the base sample $X_0$, and let $u = \hat{f}(X)$, $v = \hat{f}(Y)$ be normalized representations in $\mathbb{S}^{d-1}$, i.e., $\|u\| = \|v\| = 1$. Then*

$$\mathbb{E}_{(u,v) \sim \pi}[u \cdot v] \;\leq\; \eta_2 \;+\; (1 - \eta_2)\, \|m(\mu)\|^2, \qquad m(\mu) := \mathbb{E}[u] = \mathbb{E}[v], \tag{6}$$

*where $\eta_2 = \rho_m^2(X, X_0)$ is the squared HGR maximal correlation between the view and the base, and $\mu$ is the marginal law of $u$.*

The proof appears in Appendix A.3. This bound links the alignment of positive pairs to the structure of the statistical dependence induced by the augmentation channel. While HGR maximal correlation has been studied in statistical dependence analysis (Huang & Xu, 2020; Zhang et al., 2024), it has not previously been used to control alignment in contrastive learning. Existing work studies augmentations empirically (e.g., Tian et al. (2020)) but does not derive bounds of this form. This result formalizes how the strength of data augmentations fundamentally constrains achievable alignment under the InfoNCE objective.

## 4 GAUSSIANITY FROM INFONCE

We study why minimizing the population InfoNCE objective (Eq. 4) yields (approximately) Gaussian low-dimensional projections of learned representations, for both *normalized* representations on the sphere and *unnormalized* representations in $\mathbb{R}^d$. Our analysis proceeds along two complementary routes, which differ in the strength of the assumptions they require.

*Empirical idealization.* We first analyze an idealized regime with infinite data, ambient dimension $d \to \infty$, and sufficient optimization. Guided by empirical observations, we assume *alignment plateau* and *thin-shell concentration*; these assumptions enable a simple derivation of Gaussian projections.

*Regularized route.* To reduce reliance on training dynamics, we study a regularized variant of the population objective. Introducing a vanishing convex regularizer and assuming attainable alignment at uniformity ensures a unique minimizer and yields the same asymptotic Gaussian structure. This route provides an alternative explanation independent of training behavior.

## 4.1 GAUSSIAN PROJECTIONS AT ALIGNMENT PLATEAU

Proposition 1 provides an upper bound on achievable alignment. In the sequel we do not assume this bound is tight; instead, we model training as reaching a plateau that lies strictly below the bound.

**Assumption 1** (Alignment plateau). *After sufficient training, the positive-pair alignment saturates at a ceiling; concretely,*

$$\mathbb{E}_{(u,v)\sim\pi}[u\cdot v] \ = \ \eta_2 \ + \ r_{\text{plat}}, \tag{7}$$

*where $r_{\text{plat}} \leq 0$ is a constant error term representing the difference between the alignment value at plateau and the maximal correlation defined by the augmentations ($\eta_2$).*

Empirically, alignment saturation has been reported in some contrastive-learning settings (Wang & Isola, 2020), which motivates considering a plateau model as a plausible scenario rather than a universal requirement. In our experiments (Fig. 2, Appendix Figs. 7, 8), we frequently observe high alignment alongside improving uniformity with larger dimensions and batch sizes, suggesting that alignment may saturate before uniformity in at least some regimes. An extension that places the plateau exactly at the alignment bound (Eq. 6) is discussed in Appendix D.

**Corollary 1** (Gaussian $k$-projections at the plateau). *Suppose the alignment plateau condition (Eq. 7) holds, and consider the population objective (Eq. 4). Let $\mu^*$ denote the global minimizer supported on $\mathbb{S}^{d-1}$. Then, as $d \to \infty$, for every fixed $k \geq 1$ the $k$-dimensional marginal of $u \sim \mu^*$ satisfies*

$$\sqrt{d}\,u_k \ \Rightarrow \ \mathcal{N}(0, I_k), \tag{8}$$

*where $u_k$ denotes the projection of $u$ onto a fixed $k$-dimensional coordinate subspace and $I_k$ is the $k \times k$ identity matrix.*

The proof is provided in Appendix C.1 and follows from two lemmas. The first establishes that $\Phi(\mu)$ attains a global minimum at the uniform law (Wang & Isola, 2020), while the second invokes the central limit theorem on the sphere (Diaconis & Freedman, 1987) to deduce Gaussian projections.

### 4.1.1 GAUSSIAN PROJECTIONS FOR UNNORMALIZED REPRESENTATIONS.

So far we analyzed normalized representations on the sphere. We now extend the result to the original, unnormalized encoder outputs $z = f(X) \in \mathbb{R}^d$. Write $z = ru$, where $r = \|z\|$ is the representation radius and $u = z/\|z\| \in \mathbb{S}^{d-1}$ the normalized direction.

**Assumption 2** (Thin-shell concentration). *We assume the representation radius concentrates:*

$$\frac{r}{r_0} \ \xrightarrow[d\to\infty]{} \ 1, \tag{9}$$

*where $r_0 \in (0, \infty)$ is a deterministic constant.*

Norm concentration is widely observed in contrastive learning: unnormalized representations cluster around a characteristic radius (Wang & Isola, 2020; HaoChen et al., 2021; Levi & Gilboa, 2025). This thin-shell effect (Klartag, 2023) is further promoted by weight decay, which penalizes norm growth and stabilizes a common scale. In particular, Draganov et al. (2025) show that appropriate weight decay suppresses norm inflation and tightens the dispersion of representation norms, lending empirical support to Assumption 2. Consistent with these reports, our experiments exhibit progressively sharper radius histograms as dimension and batch size increase (Figs. 3, 4, 6).

**Proposition 2** (Gaussian projections for unnormalized representations). *Let $z = f(x) \in \mathbb{R}^d$ be the unnormalized representation and $u := z/\|z\|$. Assume $u \sim \sigma$ (the uniform distribution on $\mathbb{S}^{d-1}$) and that Assumption 2 holds, i.e., $r \xrightarrow[d\to\infty]{} r_0 \in (0, \infty)$. Then for any fixed $k$-dimensional subspace,*

$$\sqrt{d}\,z_k \ \Rightarrow \ \mathcal{N}\big(0, r_0^2 I_k\big) \qquad (d \to \infty), \tag{10}$$

*where $z_k$ denotes the orthogonal projection of $z$ onto that subspace and $I_k$ is the $k \times k$ identity.*

See proof in Appendix C.2.

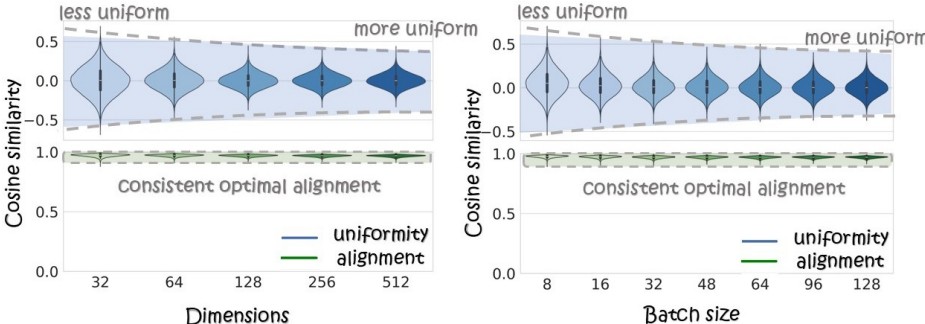

Figure 2: **Uniformity vs. alignment across settings.** A simple linear encoder trained on synthetic Laplace data exhibits (i) near-optimal alignment across all configurations and (ii) steadily improving uniformity as batch size or dimensionality grow.

## 4.2 GAUSSIAN PROJECTIONS USING REGULARIZATION

Proposition 1 shows that alignment is limited by the augmentation channel Eq. (6). At the uniform distribution ($\mu = \sigma$) the mean vanishes, $m(\sigma) = 0$, and the bound reduces to $\mathbb{E}[u \cdot v] \leq \eta_2$. Assuming this ceiling is attainable at uniformity, the uniform distribution becomes asymptotically optimal for the population objective. We work in a regularized setting, where the regularization vanishes as $d \to \infty$. As before, this has direct implications to the representation projections, which are approximately Gaussian (Theorem 2). This result shows that Gaussianity can be obtained without relying on the stronger thin-shell or plateau conditions.

We constrain $f$ to take values in $B \subseteq \mathbb{R}^d$, which is either some closed ball centered at 0 with positive radius or $\mathbb{R}^d$. We take the original loss and add two new losses: one to penalize large squared norms, and the other to encourage high entropy (we comment that both are commonly regarded as desirable goals, irrespective of our setup). Specifically, for fixed $\beta, \lambda > 0$,

$$J(f) = \Phi(\mu) - \alpha \mathbb{E}_{(u,v) \sim \pi}[u \cdot v] + \beta(-H(\rho) + \lambda \mathbb{E}_{Z \sim \rho} \|Z\|^2) , \qquad (11)$$

where $\rho = f_* p_X$ is the unnormalized pushforward probability. Define the truncated Gaussian $\gamma_\lambda^B$,

$$\gamma_\lambda^B(dz) = c_{B,\lambda} e^{-\lambda \|z\|^2} \mathbf{1}_B(z) dz , \qquad c_{B,\lambda}^{-1} = \int_B e^{-\lambda \|z\|^2} dz . \qquad (12)$$

If $\rho \ll \gamma_\lambda^B$ ($\ll$ denotes absolute continuity, so $\rho$ is absolutely continuous with respect to $\gamma_\lambda^B$), then

$$\mathrm{KL}(\rho \| \gamma_\lambda^B) = \int \log \frac{d\rho}{dz} d\rho - \int \log \frac{d\gamma_\lambda^B}{dz} d\rho = -H(\rho) + \lambda \mathbb{E}_\rho \|Z\|^2 + \log c_{B,\lambda}^{-1} , \qquad (13)$$

that is, equality up to an additive constant. Since $\rho(B) = 1$, if $\rho \not\ll \gamma_\lambda^B$, then both $\mathrm{KL}(\rho \| \gamma_\lambda^B)$ and $-H(\rho)$ are $+\infty$. Thus, it is equivalent to minimize

$$J(f) = \Phi(\mu) - \alpha \mathbb{E}_{(u,v) \sim \pi}[u \cdot v] + \beta \mathrm{KL}(\rho \| \gamma_\lambda^B) , \qquad (14)$$

and we thereby also implicitly restrict $\rho$ to satisfy $\rho \ll \gamma_\lambda^B$ and in particular $\rho(B) = 1$.

Our goal is to prove that for $\beta \geq \beta_0$, taking the angular probability as $\sigma$ approaches optimality and the optimal radial probability is that of $\gamma_\lambda^B$. If $B = \mathbb{R}^d$, this means that a Gaussian $\rho$ approaches optimality. Furthermore, as $d \to \infty$, $\beta_0 \to 0$.

This will be done in several steps. First, $\rho$ can be decomposed into a radial part and an angular part. We show that the radial part can be chosen optimally in a straightforward way.

**Proposition 3.** *Let* $\rho(dz) = \mu(du)\kappa(dr \mid u)$ *and* $\gamma_\lambda^B(dz) = \sigma(du)\xi(dr \mid u)$ *in polar coordinates* $z = ru$. *Then* $\kappa = \xi$ *is an optimal choice, yielding* $\mathrm{KL}(\rho \| \gamma_\lambda^B) = \mathrm{KL}(\mu \| \sigma)$.

The proof is given in Appendix B.1. The above proposition reduces the optimization problem for unnormalized embedding to normalized embeddings only. It also describes an optimal probability

for embedding norms, in contrast to the original InfoNCE loss, which is completely oblivious to embedding norms.

It is important to note that because we are working with a standard Borel space with a nonatomic $p_X$, any probability $\rho \in \mathcal{P}(B)$ has $\rho = g_* p_X$ for some encoding $g$. In addition, any $\mu \in \mathcal{P}(\mathbb{S}^{d-1})$ has $\mu = h_* p_X$ for some encoding, and since $B$ contains a ball around $0$, there is an encoding $f$ s.t. $h = \hat{f}$. Thus we can legitimately speak about "choosing" $\rho$ or $\mu$, since suitable encodings exist that induce them. In addition, we may also define:

**Definition 1.** *For every $\mu \in \mathcal{P}(\mathbb{S}^{d-1})$,*

$$\text{Align}(\mu) = \sup_f \left\{ \mathbb{E}[\hat{f}(X) \cdot \hat{f}(Y)] : f \text{ measurable}, \ (\hat{f})_* p_X = \mu \right\}, \tag{15}$$

As was noted, the supremum is always taken on a nonempty set. We can write

$$\tilde{J}(\mu) = \Phi(\mu) - \alpha \text{Align}(\mu) + \beta \text{KL}(\mu \| \sigma), \tag{16}$$

and it holds that $\inf_{\{\hat{f} : \hat{f}_* p_X = \mu\}} J(f) = \tilde{J}(\mu)$, and consequently $\inf_f J(f) = \inf_{\mu \in \mathcal{P}(\mathbb{S}^{d-1})} \tilde{J}(\mu)$.

The reason is that $\text{Align}(\mu)$ can be approximated arbitrarily well by an encoding, and the KL divergence is optimized by taking the radial distribution given in Proposition 3. We can therefore focus on optimizing $\tilde{J}(\mu)$.

The assumption for which we will prove our result is the following:

**Assumption 3.** *It holds that $\alpha(\eta_2 - \text{Align}(\sigma)) \xrightarrow{d \to \infty} 0$.*

We will require one more technical lemma before proceeding to prove the result.

**Lemma 1.** *If $d \geq 2$, then $\text{KL}(\mu \| \sigma) \geq C(d-1)\|m(\mu)\|^2$, where $C > 0$ is a universal constant.*

Proof is provided in Appendix B.2. To understand the constant, see (Vershynin, 2018, Proposition 2.6.1).

**Theorem 1.** *Let $d \geq 2$. There is a universal constant $C > 0$ s.t. for $\beta \geq \beta_0 = \frac{\alpha(1-\eta_2)}{C(d-1)}$,*

- *Under Assumption 3, $\tilde{J}(\sigma) - \inf_\mu \tilde{J}(\mu) \xrightarrow{d \to \infty} 0$.*

- *Assuming further that $\text{Align}(\sigma) = \eta_2$ yields that $\tilde{J}(\sigma) = \min_\mu \tilde{J}(\mu)$.*

*Moreover, as $d \to \infty$, $\beta_0 \to 0$.*

*Proof.* Write $\delta(d) = \eta_2 - \text{Align}(\sigma)$. For every $\mu$, we have that $\Phi(\mu) - \Phi(\sigma) \geq 0$ (Wang & Isola, 2020, Theorem 1). In addition,

$$\text{Align}(\mu) - \text{Align}(\sigma) \leq \eta_2 + (1 - \eta_2)\|m(\mu)\|^2 - (\eta_2 - \delta(d)) = (1 - \eta_2)\|m(\mu)\|^2 + \delta(d) \tag{17}$$

by Proposition 1. Lastly,

$$\text{KL}(\mu \| \sigma) - \text{KL}(\sigma \| \sigma) = \text{KL}(\mu \| \sigma) \geq C(d-1)\|m(\mu)\|^2 \tag{18}$$

by Lemma 1. Therefore,

$$\begin{aligned}
\tilde{J}(\mu) - \tilde{J}(\sigma) &= (\Phi(\mu) - \Phi(\sigma)) - \alpha(\text{Align}(\mu) - \text{Align}(\sigma)) + \beta(\text{KL}(\mu \| \sigma) - \text{KL}(\sigma \| \sigma)) \\
&\geq -\alpha(1 - \eta_2)\|m(\mu)\|^2 - \alpha\delta(d) + \beta C(d-1)\|m(\mu)\|^2 \\
&= (-\alpha(1 - \eta_2) + \beta C(d-1))\|m(\mu)\|^2 - \alpha\delta(d) \geq -\alpha\delta(d),
\end{aligned} \tag{19}$$

where the last inequality is by the choice of $\beta$.

If we assume that $\alpha\delta(d) \xrightarrow{d \to \infty} 0$, then $\tilde{J}(\sigma) - \inf_\mu \tilde{J}(\mu) \leq \alpha\delta(d)$, so $\tilde{J}(\sigma) - \inf_\mu \tilde{J}(\mu) \xrightarrow{d \to \infty} 0$.

If we assume further that $\text{Align}(\sigma) = \eta_2$, then $\delta(d) = 0$, and since $\tilde{J}(\sigma) \leq \tilde{J}(\mu)$ for every $\mu$, $\tilde{J}(\sigma) = \min_\mu \tilde{J}(\mu)$, completing the proof. $\qquad \square$

Since the optimal radial component of the distribution is known, we can draw conclusions w.r.t. $\rho$ as well. For example, we can directly obtain the following corollary.

**Corollary 2.** *Let $B = \mathbb{R}^d$ ($d \geq 2$) and $\beta \geq \beta_0$. If $\text{Align}(\sigma) = \eta_2$, where $\sigma$ is the uniform distribution on $\mathbb{S}^{d-1}$ and $\eta_2$ is the augmentation mildness, then $\mathcal{N}(0, (2\lambda)^{-1} I_d)$ is an optimal choice for $\rho$.*

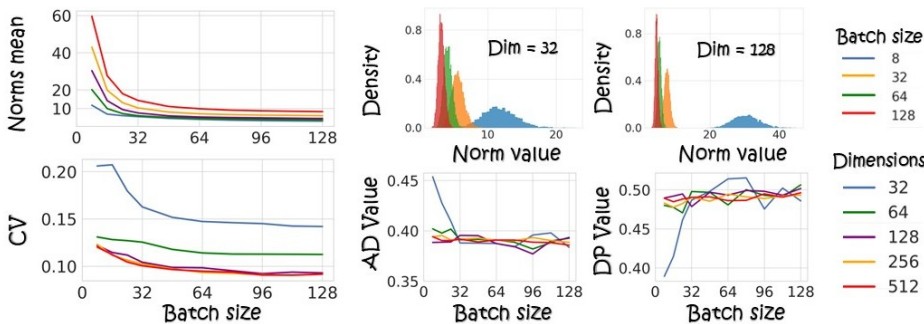

Figure 3: **Synthetic data experiments.** Left: representation norm statistics vs. batch size (curves denote dimension), showing thin-shell concentration with increasing $d$ and $N$. Top middle/right: norm histograms illustrating radius tightening. Bottom: normality diagnostics (AD, DP), with averages in the Gaussian acceptance range.

## 5 EXPERIMENTS

We empirically evaluate the distributional geometry of representations learned with the InfoNCE objective. The experiments are designed to test three theoretical predictions: (i) concentration of representation norms on a thin shell, (ii) emergence of Gaussian low-dimensional projections, and (iii) the dependence of these phenomena on contrastive learning.

We consider three settings of increasing complexity: synthetic data with linear encoders, CIFAR-10 with both contrastive and supervised training, and pretrained foundation-scale models. In all cases, we analyze both *normalized* and *unnormalized* representations. All reported trends are stable across runs; figures show representative seeds, with full implementation details in Appendix E.1.

**Metrics.** We quantify Gaussian structure using complementary diagnostics targeting radial and coordinate-wise behavior. To assess norm concentration, we measure the coefficient of variation (CV) of representation norms:

$$\text{CV} = \frac{\text{std}\big(\{\|z_i\|\}_{i=1}^N\big)}{\text{mean}\big(\{\|z_i\|\}_{i=1}^N\big)}. \tag{20}$$

$z_i$ are the learned representations and $N$ is the number of samples. A small CV indicates concentration of $\|z_i\|$ around a characteristic radius, consistent with thin-shell behavior.

To evaluate Gaussianity of low-dimensional projections, we apply two standard one-dimensional normality tests to individual coordinates: (i) the Anderson-Darling (AD) test (Anderson & Darling, 1954), where $\text{AD} < 0.752$ corresponds to failure to reject normality, and (ii) the D'Agostino-Pearson (DP) test (D'Agostino & Pearson, 1973), where $p > 0.05$ indicates failure to reject the Gaussian hypothesis. These tests probe marginal normality of fixed coordinates, as predicted by the spherical central limit theorem.

Taken together, CV captures global radial structure, while AD and DP test coordinate-level Gaussianity. This combination provides a strong finite-sample indicator of approximate Gaussian behavior and provides evidence against common heavy-tailed or mixture alternatives, which typically fail at least one of these diagnostics.

**Synthetic data experiments**. We begin with controlled synthetic settings to validate our diagnostics and isolate the mechanisms predicted by the theory. We consider three synthetic data distributions: (i) an i.i.d. Laplace$(0, 1)$ distribution, (ii) a Gaussian mixture with 25 equally weighted components and random means, and (iii) a fully discrete sparse binary distribution (1024-dimensional vectors). Each dataset contains 10k samples, and we train linear encoders using InfoNCE while varying the representation dimension and batch size. In addition, we explicitly track alignment and uniformity as functions of batch size and dimension (Fig. 2) to probe the saturation behavior predicted by our Assumption 1.

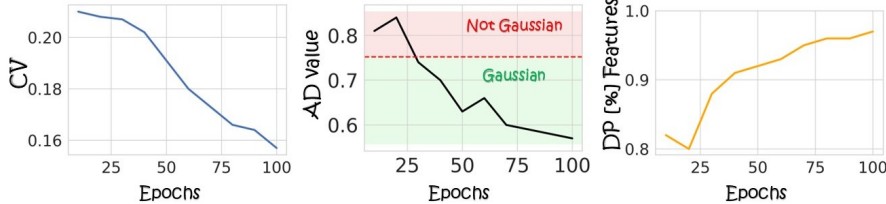

Figure 4: **CIFAR-10 training dynamics.** A two-layer MLP trained with InfoNCE on CIFAR-10 exhibits increasing Gaussianity over training. Left: representation norms concentrate as indicated by declining CV (Eq. 20). Middle: the AD statistic decreases from non-Gaussian levels into the normal range. Right: the fraction of coordinates passing the DP normality test rises steadily.

Figure 3 shows that, for Laplace inputs, representation norms progressively concentrate as both batch size and dimension increase, evidenced by a monotonic decrease in the coefficient of variation (CV). Norm histograms further illustrate the emergence of thin-shell concentration. Normality diagnostics (AD and DP) indicate that individual coordinates fall well within Gaussian acceptance thresholds, with perfect per-coordinate compliance (Table 1).

Across all three synthetic settings, including strongly non-Gaussian mixture inputs, the learned representations exhibit low norm variation and strong coordinate-wise Gaussianity (Table 1), indicating that marginal Gaussian structure emerges independently of the input distribution. The same phenomenon is observed for the fully discrete binary dataset: although representations are initially far from Gaussian, training drives pronounced norm concentration and coordinate-wise normality. Since this distribution admits no invertible mapping to a continuous Gaussian, the observed structure cannot be explained by latent Gaussian recovery.

In parallel, alignment quickly approaches a stable ceiling determined by the augmentation channel (Fig. 2), while uniformity continues to improve. This behavior is consistent with the saturation route and the emergence of isotropic Gaussian structure in high dimension. Together, these controlled experiments support the assumptions underlying our theoretical analysis and motivate the study of Gaussianity in more realistic settings.

**CIFAR-10 experiments.** We next study whether Gaussian structure emerges in a realistic vision setting. We train a two-layer MLP with a single ReLU nonlinear activation using the InfoNCE objective on CIFAR-10, and evaluate representations on the test set throughout training.

Figure 4 shows consistent trends across training: representation norms concentrate over time, as indicated by a steadily decreasing CV; the AD statistic drops from non-Gaussian levels into the normal regime; and the fraction of coordinates passing the DP test increases monotonically. These dynamics illustrate the joint emergence of thin-shell concentration and coordinate-wise Gaussianity as optimization progresses. These trends mirror the synthetic setting and show that norm concentration and Gaussianity also emerge in realistic contrastive training.

Table 1: **Gaussianity diagnostics across data and training settings.** We report norm concentration (CV) and Gaussianity via AD and DP tests (average statistic and fraction of compliant coordinates). Lower AD and higher DP indicate closer Gaussian agreement. Binary E0/E50/E100 denote evaluation at epochs 0/50/100; other results are from the end of training. Results use unnormalized embeddings.

| Metric | Synthetic (Linear) | | | | | CIFAR-10 (ResNet-18) | |
|---|---|---|---|---|---|---|---|
| | Laplace | GMM | Binary E0 | E50 | E100 | Supervised | Contrastive |
| CV | 0.08 | 0.08 | 0.36 | 0.12 | 0.09 | 0.5 | 0.09 |
| AD Avg. ($< 0.752$) | 0.38 | 0.39 | 1.64 | 0.41 | 0.42 | 3.3 | 0.43 |
| AD Norm. Feat. | 100% | 100% | 30% | 93% | 97% | 6.2% | 96.1% |
| DP Avg. ($> 0.05$) | 0.49 | 0.46 | 0.02 | 0.44 | 0.46 | 0.041 | 0.39 |
| DP Norm. Feat. | 100% | 100% | 15% | 89% | 98% | 3.9% | 94.5% |
| **Gaussian?** | ✓ | ✓ | ✗ | ✓ | ✓ | ✗ | ✓ |

**Contrastive vs. supervised training.** To isolate the role of the training objective, we compare supervised and contrastive learning using the same ResNet-18 architecture on CIFAR-10, following the SimCLR training protocol. Both models share identical initialization and capacity, differing only in the objective: cross-entropy supervision versus InfoNCE. As shown in Table 1, supervised training yields representations with high norm variability and strong deviations from Gaussianity, with most coordinates failing both AD and DP tests. In contrast, InfoNCE training produces concentrated norms and near-Gaussian per-coordinate distributions. These results indicate that the emergence of Gaussian structure is not explained by the data or architecture alone, but is also a direct consequence of the contrastive objective.

Table 2: **Gaussianity diagnostics for pretrained models.** Coordinate-wise Gaussianity via AD and DP tests (average statistic and fraction of compliant coordinates). Test thresholds are indicated in headers. Results shown for *Unnormalized / Normalized* embeddings.

| Training | Model (Test data) | AD ($< 0.752$) | | DP ($> 0.05$) | | Gaussian? |
|---|---|---|---|---|---|---|
| | | Avg. | Norm. Feat. (%) | Avg. | Norm. Feat. (%) | |
| Supervised | ResNet-34 (MS-COCO) | 10.01 / 9.638 | 0.0% / 0.0% | $2.2 \times 10^{-6}$ / $3.2 \times 10^{-6}$ | 0.0% / 0.0% | ✗ |
| | DenseNet (MS-COCO) | 2.982 / 2.8538 | 42.2% / 41.6% | 0.1550 / 0.1442 | 49.3% / 49.0% | ✗ |
| Self-supervised | DINO (MS-COCO) | 0.44 / 0.44 | 97.0% / 97.1% | 0.45 / 0.45 | 99.2% / 99.3% | ✓ |
| | CLIP (Image, MS-COCO) | 0.47 / 0.49 | 96.8% / 96.0% | 0.42 / 0.39 | 99.6% / 99.4% | ✓ |
| | CLIP (Text, MS-COCO) | 0.53 / 0.54 | 94.0% / 93.6% | 0.38 / 0.38 | 99.4% / 99.7% | ✓ |
| | CLIP (Image, Sketch) | 0.4 / 0.4 | 94.8% / 94.7% | 0.44 / 0.42 | 93.3% / 93.2% | ✓ |
| | CLIP (Image, Painting) | 0.41 / 0.42 | 95.3% / 95.1% | 0.43 / 0.4 | 94.2% / 93.9% | ✓ |

**Pretrained models.** We further examine whether Gaussian structure persists in large pretrained representations. On the MS-COCO validation set (Lin et al., 2014), we compare self-supervised backbones CLIP (ViT-L/14 image and text encoders) (Radford et al., 2021) and DINO (ViT-B/32) (Caron et al., 2021) against supervised ImageNet-pretrained (Deng et al., 2009), ResNet34 (He et al., 2016), and DenseNet (Huang et al., 2017). Normality diagnostics (Table 2) show that self-supervised models exhibit near-Gaussian coordinate distributions, while supervised models deviate substantially. We further evaluate the CLIP image encoder on ImageNet-R (Sketch and Painting domains) to test robustness beyond natural images, and again observe strong Gaussian signatures. Although CLIP and DINO are not exact instances of the unimodal InfoNCE setting, these diagnostics suggest that similar isotropic Gaussian-like statistics may arise broadly in self-supervised objectives.

## 6 DISCUSSION AND CONCLUSION

We showed that InfoNCE trained representations admit an asymptotic Gaussian law, via two routes: an alignment-plateau analysis with thin-shell concentration, and a regularized surrogate with milder assumptions. Experiments on synthetic data, CIFAR-10, and pretrained models (MS-COCO and ImageNet-R) are consistent with these assumptions and the Gaussian hypothesis, revealing norm concentration, alignment saturation, and near-Gaussian projections. These results indicate that the Gaussian convergence remains informative well before the infinite-dimensional limit. This Gaussian view justifies common modeling choices (e.g., likelihood scoring, OOD detection) and suggests that explicit isotropy promoting regularizers may act as principled surrogates for InfoNCE's implicit bias. However, *limitations* remain: our results are asymptotic, relying on high-dimensional limits and idealized assumptions that may not capture all practical regimes. We therefore view our asymptotic framework as a principled starting point rather than a complete description of all practical regimes. For finite dimension $d$ and batch size $N$, projections are close to Gaussian, with deviations vanishing as $d, N \to \infty$. Quantitative bounds follow from classical Berry-Esseen (Vershynin, 2018) rates in high dimension and uniform laws of large numbers for empirical objectives (Wellner et al., 2013). In particular, the minimizer of the empirical InfoNCE loss deviates from the population minimizer by $O(N^{-1/2})$ according to Wang & Isola (2020, Thm. 1), and the distribution of fixed-$k$ projections deviates from Gaussian by $O(d^{-1})$ according to Diaconis & Freedman (1987) (see Theorem 2 in Appendix C.1). Thus, for large but finite $d, N$, the Gaussian limit provides a representative and empirically useful approximation. In addition, we do not analyze optimization dynamics or prove that training attains these minimizers in practice; our results are asymptotic and characterize the population optima under the stated assumptions. Overall, we provide a principled asymptotic explanation for Gaussianity in contrastive representations, grounding empirical observations and opening new directions for analysis and practical design.

ETHICS STATEMENT

This work is theoretical and empirical in nature, focused on understanding the statistical behavior of representations trained with contrastive learning. We do not foresee direct negative societal impacts. Potential downstream applications of Gaussian modeling (e.g., density estimation, OOD detection) could influence decisions in safety-critical domains, and care must be taken to ensure robustness and fairness.

REPRODUCIBILITY

We provide detailed descriptions of theoretical assumptions, proofs, and experimental protocols. Datasets (synthetic data, CIFAR-10 (Krizhevsky et al., 2009), and MS-COCO (Lin et al., 2014)) are publicly available. Architectures, hyperparameters, and training settings are fully specified (Appendix E.1), and to ensure reproducibility, code for the experiments is released here.

ACKNOWLEDGMENTS

We would like to acknowledge support by the Israel Science Foundation (Grant 1472/23) and by the Ministry of Innovation, Science and Technology (Grants No. 5074/22, 8801/25).

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

## LLM USAGE

Portions of this manuscript, including text editing, reference search, ideation, mathematical derivations, and summarization, were assisted by a large language model. The model was used interactively to refine exposition, suggest formulations, and check consistency of notation, but all results, proofs, and experiments were implemented and validated by the authors. All mathematical claims, experimental details, and citations were independently verified. No content was included without author review and approval.

## OVERVIEW

This appendix provides complete proofs for all propositions, corollaries, lemmas, and theorems, along with additional derivations that did not fit in the main text. We also include supplementary experiments and implementation details. The appendices are organized as follows:

A. Proof and details of the alignment bound (Sec. A).

B. Proofs of some regularization surrogate-related claims (Sec. B).

C. Proof of the alignment-plateau approach. These include general claims, some are used in the regularization surrogate proof as well (Sec. C).

D. Discussion about exact alignment bound at plateau (Sec. D).

E. Experimental details (Sec. E).

## A HGR MAXIMAL CORRELATION AND THE ALIGNMENT BOUND

### A.1 HGR DEFINITION AND BASIC PROPERTIES

The Hirschfeld-Gebelein-Rényi (HGR) maximal correlation (Hirschfeld, 1935; Gebelein, 1941; Rényi, 1959) between random variables $X$ and $Y$ is

$$\rho_m(X, Y) := \sup_{\substack{\mathbb{E}[\phi(X)]=\mathbb{E}[\psi(Y)]=0 \\ \mathrm{Var}(\phi)=\mathrm{Var}(\psi)=1}} \mathbb{E}[\phi(X)\psi(Y)] \in [0, 1]. \tag{21}$$

An equivalent "explained-variance" characterization (Gebelein, 1941; Rényi, 1959) is

$$\rho_m^2(X, Y) = \sup_{\substack{g \in L^2(p_X) \\ \mathrm{Var}(g(X))>0}} \frac{\mathrm{Var}(\mathbb{E}[g(X) \mid Y])}{\mathrm{Var}(g(X))}. \tag{22}$$

Here $p_X$ is the marginal law of $X$, and $L^2(p_X)$ denotes the square-integrable (measurable) functions of $X$ under $p_X$. The numerator is the variance explained by the optimal $L^2$ predictor $\mathbb{E}[g(X) \mid Y]$ and the denominator is its total variance. Hence, the ratio is a (generalized) coefficient of determination, i.e., the fraction of variance of $g(X)$ predictable from $Y$, in $[0, 1]$.

HGR satisfies a (multiplicative) data-processing inequality (DPI): if $X - Y - Z$ is a Markov chain, then

$$\rho_m(X, Z) \leq \rho_m(X, Y)\,\rho_m(Y, Z) \quad \text{(Rényi, 1959)}. \tag{23}$$

Our representations are normalized ($u, v \in \mathbb{S}^{d-1}$), hence bounded and each coordinate is in $L^2$. See also the operator-theoretic interpretation in (Anantharam et al., 2013).

### A.2 GAUSSIAN EXAMPLE

If two random variables $X$ and $Y$ are jointly Gaussian, then the HGR maximal correlation between them equals the absolute value of their Pearson correlation coefficient:

$$\rho_m(X, Y) = |A|, \qquad A := \frac{\mathrm{Cov}(X, Y)}{\sqrt{\mathrm{Var}(X)\mathrm{Var}(Y)}}. \tag{24}$$

This is a special case where the supremum in the HGR definition is achieved by simple linear functions. More precisely, the optimal transformations are just standardized versions of $X$ and $Y$ themselves. In other words, nonlinear functions cannot increase correlation beyond the linear one when the joint distribution is Gaussian. This result is well established; see, for example, Bryc & Dembo (2005).

### A.3 PROOF OF THE ALIGNMENT BOUND

We prove the inequality

$$\mathbb{E}[u \cdot v] \leq \eta_2 + (1 - \eta_2)\,\|m(\mu)\|^2, \tag{25}$$

for normalized representations $u = \hat{f}(X)$ and $v = \hat{f}(Y)$ on $\mathbb{S}^{d-1}$, where $m(\mu) := \mathbb{E}[u] = \mathbb{E}[v]$ is their common mean.

**Step 1: mean-residual decomposition.** Since $u$ and $v$ share the same marginal $\mu$, their means coincide:

$$m(\mu) := \mathbb{E}[u] = \mathbb{E}[v]. \tag{26}$$

Define residuals

$$\tilde{u} := u - m(\mu), \qquad \tilde{v} := v - m(\mu), \tag{27}$$

so that $\mathbb{E}[\tilde{u}] = \mathbb{E}[\tilde{v}] = 0$. Expanding the inner product yields

$$\mathbb{E}[u \cdot v] = \mathbb{E}\big[(m(\mu) + \tilde{u}) \cdot (m(\mu) + \tilde{v})\big] = \|m(\mu)\|^2 + \mathbb{E}[\tilde{u} \cdot \tilde{v}]. \tag{28}$$

The cross terms vanish because $\mathbb{E}[\tilde{u}] = \mathbb{E}[\tilde{v}] = 0$, so

$$\mathbb{E}[m(\mu) \cdot \tilde{v}] = m(\mu) \cdot \mathbb{E}[\tilde{v}] = 0, \tag{29}$$

and

$$\mathbb{E}[\tilde{u} \cdot m(\mu)] = \mathbb{E}[\tilde{u}] \cdot m(\mu) = 0. \tag{30}$$

**Step 2: bound the residual correlation via HGR.** Fix a coordinate $k \in \{1, \ldots, d\}$ and set

$$g_k(X) := \tilde{u}_k, \qquad h_k(Y) := \tilde{v}_k. \tag{31}$$

Then $\mathbb{E}[g_k(X)] = \mathbb{E}[h_k(Y)] = 0$ and, by the Markov structure $X - X_0 - Y$ the DPI for HGR maximal correlation gives

$$\rho_m(X, Y) \leq \rho_m(X, X_0) \rho_m(X_0, Y) = \sqrt{\eta_2} \sqrt{\eta_2} = \eta_2, \tag{32}$$

as in Anantharam et al. (2013).

For any real-valued, square-integrable functions $g(X)$, $h(Y)$ with zero mean, we can apply the definition of HGR maximal correlation from Eq. (21) together with the Cauchy-Schwarz inequality to obtain:

$$\left| \mathbb{E}[g(X) h(Y)] \right| \leq \rho_m(X, Y) \sqrt{\mathrm{Var}(g) \mathrm{Var}(h)}. \tag{33}$$

This inequality holds even when $g$ and $h$ are not normalized, since any such functions can be rescaled to have unit variance. In our case, the random variables $X$ and $Y$ are conditionally independent given $X_0$, and identically drawn from the same augmentation channel $\mathcal{A}(\cdot \mid X_0)$. Therefore, the Markov chain $X \leftarrow X_0 \rightarrow Y$ holds, and the multiplicative data-processing inequality (Eq. 32) gives:

$$\rho_m(X, Y) \leq \rho_m(X, X_0) \rho_m(Y, X_0) = \eta_2. \tag{34}$$

Substituting Eq. (34) into Eq. (33) yields:

$$\left| \mathbb{E}[g(X) h(Y)] \right| \leq \eta_2 \sqrt{\mathrm{Var}(g) \mathrm{Var}(h)}. \tag{35}$$

Applying Eq. (35) to $(g_k, h_k)$ and summing over coordinates,

$$\mathbb{E}[\tilde{u} \cdot \tilde{v}] = \sum_{k=1}^{d} \mathbb{E}[\tilde{u}_k \tilde{v}_k] \leq \eta_2 \sum_{k=1}^{d} \sqrt{\mathrm{Var}(\tilde{u}_k) \mathrm{Var}(\tilde{v}_k)} \leq \eta_2 \sqrt{\sum_{k=1}^{d} \mathrm{Var}(\tilde{u}_k)} \sqrt{\sum_{k=1}^{d} \mathrm{Var}(\tilde{v}_k)}, \tag{36}$$

where the last step is Cauchy-Schwarz for sequences.

**Step 3: compute the marginal variances.** Because $\|u\| = \|v\| = 1$ and $m(\mu) = \mathbb{E}[u] = \mathbb{E}[v]$,

$$\sum_{k=1}^{d} \mathrm{Var}(\tilde{u}_k) = \mathbb{E}[\|\tilde{u}\|^2] = \mathbb{E}[\|u - m(\mu)\|^2] = \mathbb{E}[\|u\|^2] - \|m(\mu)\|^2 = 1 - \|m(\mu)\|^2, \tag{37}$$

and identically

$$\sum_{k=1}^{d} \mathrm{Var}(\tilde{v}_k) = 1 - \|m(\mu)\|^2. \tag{38}$$

**Step 4: conclude.** Combine Eq. (36), Eq. (37) and Eq. (38) to get

$$\mathbb{E}[\tilde{u} \cdot \tilde{v}] \leq \eta_2 \left( 1 - \|m(\mu)\|^2 \right). \tag{39}$$

## B REGULARIZED SURROGATE PROOFS

### B.1 PROOF OF PROPOSITION 3

*Proof.* For any encoder $f$ with angular law $\mu$ the KL term satisfies (by the KL chain rule, see e.g. Dupuis & Ellis, 2011, Theorem B.2.1)

$$\mathrm{KL}(\rho\|\gamma_\lambda^B) = \mathrm{KL}(\mu\|\sigma) + \int \mathrm{KL}\big(\kappa(\cdot \mid u) \,\|\, \xi(\cdot \mid u)\big) \mu(du), \tag{40}$$

where $\rho(dz) = \mu(du)\kappa(dr \mid u)$ and $\gamma_\lambda^B(dz) = \sigma(du)\xi(dr \mid u)$ in polar coordinates $z = ru$. Thus, at fixed $\mu$, the KL term is minimized by choosing $\kappa(\cdot \mid u) = \xi(\cdot \mid u)$ $\mu$-a.s., and then $\mathrm{KL}(\rho\|\gamma_\lambda^B) = \mathrm{KL}(\mu\|\sigma)$. $\qquad \square$

## B.2 PROOF OF LEMMA 1

*Proof.* We can assume $\mu \ll \sigma$, otherwise $\mathrm{KL}(\mu\|\sigma) = +\infty$ and the claim is trivial. The claim is also trivially true if $m(\mu) = 0$, so assume $m(\mu) \neq 0$. By the Donsker-Varadhan variational formula (Dupuis & Ellis, 2011, Lemma 1.4.3)

$$\mathrm{KL}(\mu\|\sigma) = \sup_{\varphi} \left\{ \mathbb{E}_{u\sim\mu}[\varphi(u)] - \log \mathbb{E}_{u\sim\sigma}\left[e^{\varphi(u)}\right] \right\}, \tag{41}$$

where the supremum is taken over bounded measurable functions $\varphi : \mathbb{S}^{d-1} \to \mathbb{R}$. Taking $\varphi(u) = tw \cdot u$ for some unit vector $w \in \mathbb{R}^d$ and $t \in \mathbb{R}$, we have

$$\mathrm{KL}(\mu\|\sigma) \geq \mathbb{E}_{u\sim\mu}[tw \cdot u] - \log \mathbb{E}_{u\sim\sigma}\left[e^{tw\cdot u}\right] = tw \cdot m(\mu) - \log \mathbb{E}_{u\sim\sigma}\left[e^{tw\cdot u}\right]. \tag{42}$$

Suppose we showed that

$$\log \mathbb{E}_{u\sim\sigma}\left[e^{tw\cdot u}\right] \leq t^2/a \tag{43}$$

for some $a > 0$ for every choice of $t$ and $w$. Then picking $t = \frac{a}{2}\|m(\mu)\|$ and $w = m(\mu)/\|m(\mu)\|$, we have

$$\mathrm{KL}(\mu\|\sigma) \geq tw \cdot m(\mu) - t^2/a = \frac{a}{2}\|m(\mu)\|^2 - \frac{a}{4}\|m(\mu)\|^2 = \frac{a}{4}\|m(\mu)\|^2. \tag{44}$$

It is left to show Eq. (43) with $a = 4C(d-1)$. Now, since $g(u) = w \cdot u$ is 1-Lipschitz on the sphere, then by a corollary of Lévy's isoperimetric inequality, for all $s \geq 0$,

$$\sigma\left(|g| \geq s\right) \leq 2e^{-\frac{1}{2}(d-1)s^2}, \tag{45}$$

where we used the fact that the median of $g$ is 0. Since $\mathbb{E}g = 0$, this implies that for some universal $C' > 0$,

$$\log \mathbb{E}e^{tg} \leq \frac{2C'^2 t^2}{d-1} \tag{46}$$

(Vershynin, 2018, Proposition 2.6.1). This satisfies Eq. (43) with $a = \frac{d-1}{2C'^2}$, and taking $C = 1/(8C'^2)$, we are done. $\qquad\square$

## C ALIGNMENT-PLATEAU PROOFS

### C.1 NORMALIZED REPRESENTATIONS

**Lemma 2** (At the plateau the loss reduces to uniformity). *Under Assumption 1, the population InfoNCE objective (Eq. 4) takes the form*

$$\mathcal{J}(\mu) = \Phi(\mu) - \alpha \mathbb{E}[u \cdot v] = \Phi(\mu) - \alpha\big(\eta_2 + r_{\mathrm{plat}}\big). \tag{47}$$

*hence minimizing $\mathcal{J}$ over probability laws $\mu$ on $\mathbb{S}^{d-1}$ is equivalent to minimizing $\Phi(\mu)$. Moreover, $\Phi(\mu)$ is uniquely minimized by the uniform law $\sigma$ on $\mathbb{S}^{d-1}$.*

*Proof.* At the plateau, $\mathbb{E}[u \cdot v]$ is the constant in Eq. (7), so the alignment term is independent of $\mu$, leaving the uniformity potential $\Phi(\mu)$ as the only objective. By Wang & Isola (2020, Appendix A), $\Phi$ is uniquely minimized at the uniform distribution on the sphere, i.e. $\mu = \sigma$. For consistency, the plateau value in Eq. (7) must be feasible at $\mu = \sigma$. $\qquad\square$

*Remark.* In Eq. (7), $r_{\mathrm{plat}} \leq 0$. By Eq. (6) at $\mu = \sigma$ ($m(\mu) = 0$) the alignment ceiling is $\eta_2$; the plateau value is not guaranteed to be feasible at $\mu = \sigma$ and must be verified.

**Lemma 3** (Maxwell-Poincaré (Diaconis & Freedman, 1984)). *Let $U_d$ be uniform on $\mathbb{S}^{d-1}$, where $U_{d,i}$ denotes the $i$-th coordinate of $U_d$. Fix $k \in \mathbb{N}$. Then*

$$\sqrt{d}(U_{d,1}, \ldots, U_{d,k}) \Rightarrow \mathcal{N}(0, I_k) \qquad (d \to \infty). \tag{48}$$

A concrete rate of convergence was given by Diaconis & Freedman (1987).

**Theorem 2.** *(Diaconis & Freedman, 1987) If $1 \leq k \leq d - 4$, then*

$$d_{\mathrm{TV}}\big(\sqrt{d}(U_{d,1}, \ldots, U_{d,k}), Z\big) \leq \frac{2(k+3)}{d-k-3}, \tag{49}$$

*where $Z \sim \mathcal{N}(0, I_k)$ and $d_{\mathrm{TV}}$ denotes the total variation distance.*

Clearly, Lemma 3 and Theorem 2 hold for any $k$ indices, or for any orthonormal projection of $U_d$ to $k$ dimensions. Combining Lemmas 2 and 3, we get Corollary 1.

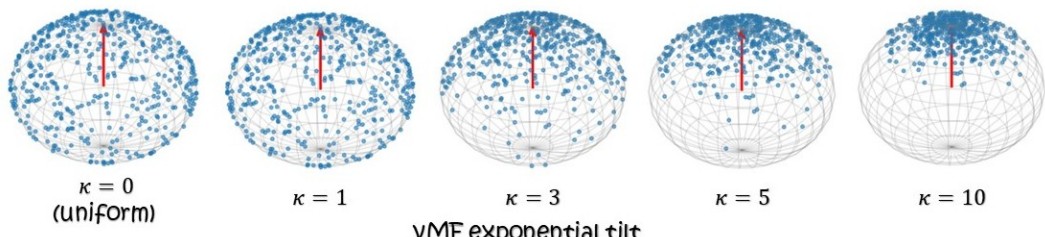

Figure 5: **vMF exponential tilt distribution for different concentration scales kappa ($\kappa$).**

## C.2 UNNORMALIZED REPRESENTATIONS

We now prove Proposition 2 by reducing to the normalized case established above.

*Proof.* Let $z = f(X) \in \mathbb{R}^d$ denote the unnormalized representation and write its polar decomposition as $z = r\,u$ with $r = \|z\| > 0$ and $u := z/\|z\| \in \mathbb{S}^{d-1}$. By Lemma 2, at the alignment plateau the population objective reduces to minimizing $\Phi(\mu)$, whose unique minimizer is the uniform law $\sigma$ on $\mathbb{S}^{d-1}$. Hence the angular component of any global minimizer satisfies $u \sim \sigma$ on $\mathbb{S}^{d-1}$. Assumption 2 further gives thin-shell concentration of the radius: $r \xrightarrow[d\to\infty]{\mathsf{P}} r_0 \in (0, \infty)$.

For any fixed $k \geq 1$ and any fixed $k$-dimensional subspace, let $P_k$ be the corresponding orthogonal projector and set $u_k := P_k u$. By the Maxwell-Poincaré spherical CLT (Lemma 3),

$$\sqrt{d}\,u_k \;\Rightarrow\; \mathcal{N}(0, I_k) \qquad (d \to \infty). \tag{50}$$

Let $z_k := P_k z = r\,u_k$. Since $r \xrightarrow[d\to\infty]{\mathsf{P}} r_0$ and Eq. (50) holds, Slutsky's theorem (Van der Vaart, 2000) yields

$$\sqrt{d}\,z_k \;=\; r\sqrt{d}\,u_k \;\Rightarrow\; \mathcal{N}(0, r_0^2 I_k) \qquad (d \to \infty). \tag{51}$$

This proves Proposition 2. $\qquad\square$

## D EXACT ALIGNMENT BOUND IN PLATEAU DISCUSSION

The following analysis begins from the alignment ceiling (Eq. 6): under a generalized plateau assumption (extending Assumption 1), the expected alignment is determined by the augmentation mildness $\eta_2$ and the squared mean norm $\|m(\mu)\|^2$, up to a negligible residual (noted as $r_{plat}$ in Eq. (7)). Substituting this relation into the population InfoNCE objective (Eq. 4) yields the surrogate

$$\mathcal{J}_q(\mu) \;=\; \Phi(\mu) \;-\; q\,\|m(\mu)\|^2, \qquad q = \alpha(1-\eta_2), \tag{52}$$

where $\Phi(\mu)$ is the uniformity potential of Wang & Isola (2020). Thus, at the plateau, the population loss reduces to a trade-off between uniformity and the mean vector length.

**Stationary points.** The surrogate involves the spherical convolution operator $P$ with kernel $e^{\alpha\xi\cdot\eta}$, which diagonalizes in spherical harmonics by the Funk-Hecke theorem (Atkinson & Han, 2012). Analyzing the Euler-Lagrange condition shows that in high dimensions $Ph$ must asymptotically take an exponential tilt form $Ph(\xi) \propto \exp(\beta w \cdot \xi)$. Inverting this relation via Gegenbauer expansions and their decay properties (Szeg, 1939) indicates that, under mild regularity, the stationary density $h$ is well-approximated in its leading modes by either the uniform law or a von Mises-Fisher (vMF) tilt (Mardia & Jupp, 2009). This captures the dominant low-degree structure in high dimensions, though more complex stationary forms cannot be excluded.

**Implications.** Consequently, in high dimension the stationary points of the plateau surrogate are *well-approximated* by either the uniform distribution (when $m(\mu) = 0$) or a von Mises-Fisher (vMF) tilt aligned with an axis $w$ (when $m(\mu) \neq 0$); see Fig. 5. The vMF concentration parameter $\kappa$ quantifies the strength of angular concentration around $w$ (larger $\kappa \Rightarrow$ narrower cone). This

perspective helps explain why contrastive encoders often yield nearly uniform representations, with occasional vMF-like bias. For example, in CLIP, where a narrow-cone structure (a modality-dependent angular bias) has been observed (Liang et al., 2022).

# E EXPERIMENTAL DETAILS

## E.1 IMPLEMENTATION DETAILS

**Code and reproducibility.**   Code is available here. All experiments were implemented in `PyTorch` with `torchvision`. Training was performed on a single 3090 NVIDIA RTX GPU with CUDA 11.8.

**Synthetic Laplace data experiments.**

- **Dataset.** Laplace$(0, 1)$ vectors of dimensions - $d_{data} = 1024$. We use a set of 20k samples for training, and 5k samples for testing.
- **Representation dimensions.**   The dimensions of representations vary:   $d \in \{32, 64, 128, 256\}$.
- **Batch size.** Batch size in our experiments varies: $N \in \{8, 16, 32, 48, 64, 96, 128\}$.
- **Training objective.** InfoNCE loss with temperature $\tau \in \{0.1, 0.2\}$ . We report results for $\tau = 0.1$, but note that results are similar.
- **Augmentations.** Each synthetic sample $x$ is perturbed to form two correlated views

$$x_1 = A\,x + \sqrt{1 - A^2}\,\varepsilon_1, \qquad x_2 = A\,x + \sqrt{1 - A^2}\,\varepsilon_2, \tag{53}$$

  where $\varepsilon_1, \varepsilon_2 \sim \mathcal{N}(0, I)$ are independent. The parameter $A \in (0, 1)$ controls the correlation between views. After this linear Gaussian mixing, we apply light, independent jitter: additive Gaussian noise with std $0.2$, feature dropout with probability $0.1$, and random multiplicative scaling by $\exp(\mathcal{N}(0, 0.1^2))$. Unless otherwise stated, we use $A = 0.6$ (results for $A \in \{0.2, 0.5, 0.8\}$ appear in Fig. 11).
- **Optimization.** Optimizer: Adam. Learning rate $= 10^{-3}$. We ran 50-250 epochs depending on setup; unless stated otherwise, we report results at 150 epochs.
- **Evaluation metrics.** norm concentration (CV), mean norm values, Gaussianity diagnostics (AD/DP) tests and uniformity vs. alignment comparison (based on cosine similarity).

**Additional synthetic data experiments.**

For the Gaussian mixture setting:

- **Dataset.** We generate 10k samples in $\mathbb{R}^d$ from a mixture of 25 equally weighted Gaussian components (1024 dimensions) with randomly sampled means and shared isotropic covariance. The component means are drawn independently at initialization and fixed throughout training.
- **Augmentation.** Positive pairs are generated by independently sampling two views from the same underlying mixture component.
- **Training.** A linear encoder maps inputs to a 256-dimensional representation space and is trained using the InfoNCE objective for 100 epochs.
- **Evaluation metrics.**   Normality diagnostics include norm concentration (CV) and coordinate-wise AD and DP statistics.

For the discrete binary setting:

- **Dataset.** Each sample is a sparse binary vector of dimension 1024.
- **Augmentation.** Positive pairs are generated by independently flipping a small fraction (0.1%) of zero entries to ones.

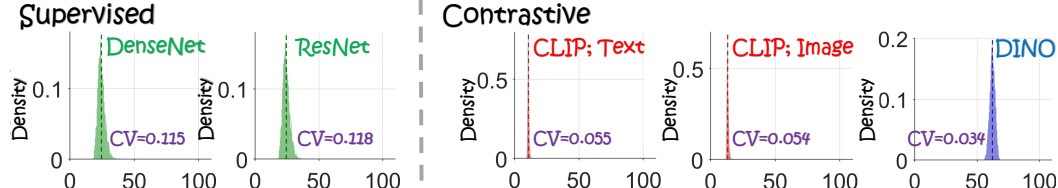

Figure 6: **Thin-shell concentration across pretrained models.** Radius distributions of representations from supervised models (DenseNet, ResNet) and contrastive models (CLIP, DINO). All models exhibit thin-shell concentration, with contrastive methods showing tighter clustering (lower CV, Eq. (20)).

- **Training.** A linear encoder maps inputs to a 256-dimensional representation and is trained using the InfoNCE objective for 100 epochs.

- **Evaluation metrics.** Every 10 epochs, we evaluate normality diagnostics, including norm concentration (CV) and coordinate-wise AD and DP statistics.

**CIFAR-10 experiments**

- **Dataset.** CIFAR-10, training set size 50k, test set size 10k.

- **Augmentations.** We apply the standard SimCLR-style augmentation pipeline: a random resized crop to $32 \times 32$ pixels with scale uniformly sampled from $(0.2, 1.0)$, a random horizontal flip, color jitter with strengths $(0.8, 0.8, 0.8, 0.2)$, and random conversion to grayscale with probability $0.2$.

- **Architecture.** Two experimental settings: (i) a basic encoder composed of a two-layer MLP with nonlinearity, and (ii) a ResNet-18 encoder following the SimCLR protocol for the contrastive setting and trained with standard cross-entropy for the supervised setting (pretrained on ImageNet (Deng et al., 2009)).

- **Training objective.** InfoNCE with temperature $\tau = 0.1$ (cross-entropy for the supervised setting).

- **Optimization.** Adam optimizer, learning rate $= 10^{-3}$, weight decay $= 10^{-4}$, batch size $= 256$, epochs $= 100$.

- **Evaluation metrics.** norm concentration (CV), Gaussianity diagnostics (AD/DP) tests.

**Pretrained model diagnostics**

- **Models.** CLIP (ViT-L/14, text and image modalities), DINO (ViT-B/32), ResNet-34 and DenseNet.

- **Datasets.** Full MS-COCO validation set (5k images) and the full ImageNet-R benchmark when noted.

- **Feature extraction.** Last-layer embeddings; whitening applied when noted.

- **Evaluation metrics.** norm concentration (CV), Gaussianity diagnostics (AD/DP) tests and uniformity before and after whitening.

## E.2 ADDITIONAL EXPERIMENTS

**Thin-shell concentration in pretrained models.** Fig. 6 visualizes the radius distributions of representations from supervised (DenseNet, ResNet34) and self-supervised (CLIP image/text, DINO) pretrained models on MS-COCO. All models exhibit thin-shell concentration, with radius values tightly clustered around a characteristic norm. Notably, contrastive models display significantly stronger concentration (lower CV) than supervised counterparts, consistent with the Gaussian diagnostics reported in Table 2. This reinforces the underlying near-Gaussian structure observed in self-supervised representations.

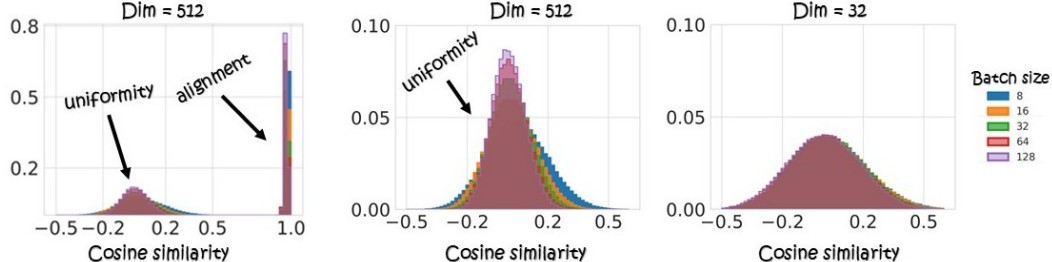

Figure 7: **Alignment and uniformity vs. batch size.** Histogram view of cosine similarities for positive pairs (alignment) and negatives (uniformity), corresponding to Fig. 2. As batch size increases, alignment remains high while uniformity improves, with negative-pair similarities concentrating near zero. The middle panel is a zoom of the left; the right panel shows that at very low dimensionality, increasing batch size yields little uniformity gain.

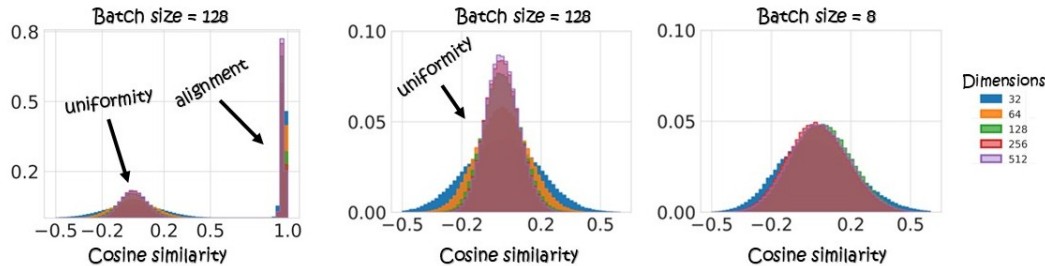

Figure 8: **Alignment and uniformity vs. dimensionality.** Histogram view of cosine similarities for positive pairs (alignment) and negatives (uniformity), corresponding to Fig. 2. As dimensionality increases, alignment stays high while uniformity improves, pushing negative-pair similarities toward zero. The middle panel is a zoom of the left; the right panel highlights that with very small batch sizes, increasing dimensionality offers limited uniformity improvement.

Figs. 7 and 8 provide alternative visualizations of Fig. 2, presenting the same experiments with a different display. Both figures plot the distributions of cosine similarities for positive pairs (alignment) and for negatives (uniformity). As batch size (Fig. 7) or dimensionality (Fig. 8) increases, uniformity improves (negative-pair similarities concentrate near zero) while alignment remains consistently high across settings. These complementary views reinforce the observation from the main body: uniformity continues to improve with larger batches and higher dimensions, whereas alignment appears to saturate early.

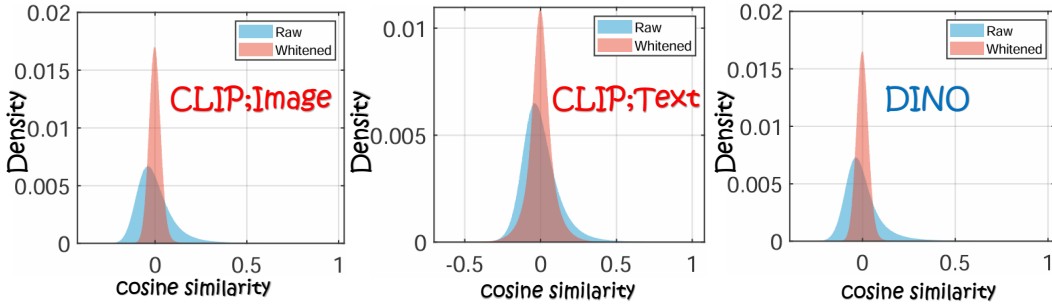

Figure 9: **Whitening and uniformity: unnormalized representations.** Cosine similarity histograms of negatives for CLIP (image, text) and DINO, before (raw) and after whitening. Unnormalized representations benefit from whitening, with distributions pushed closer to zero, reflecting enhanced uniformity. The y-axis ("Density") represents the relative count in each bin (normalized by the total number of samples) rather than the probability density function.

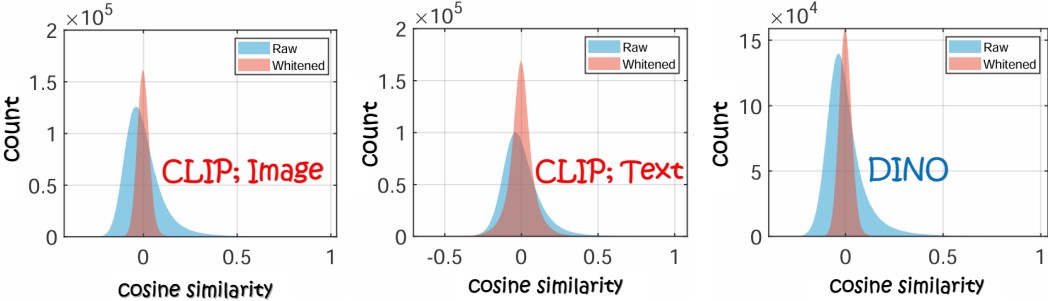

Figure 10: **Whitening and uniformity: normalized representations.** Cosine similarity histograms of negatives for CLIP (image, text) and DINO, before (normalized) and after whitening. Normalized representations are already close to uniform; whitening provides a modest but consistent improvement.

Additionally, we assess uniformity in several pretrained models before and after whitening. Whitening consistently increases uniformity, indicating that these representations, which are already close to uniform (and approximately Gaussian; see Table 2), become more isotropic once decorrelated and rescaled. This effect holds consistently across pretrained models (CLIP image, CLIP text, and DINO), for both normalized and unnormalized representations, see Figs. 9, 10. Thus, a simple post hoc projection via whitening can further enhance uniformity in practice.

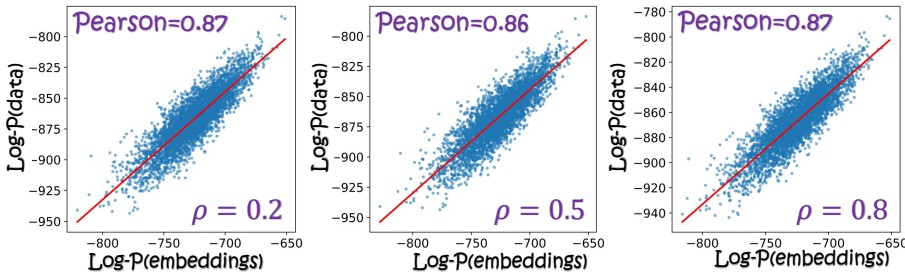

Figure 11: **Encoder "pushforward".** On synthetic data, the encoder maps Laplace-distributed inputs to approximately Gaussian representations. Because both source and target families admit tractable likelihoods, we can score entire sets and observe consistently high correlation across different augmentation strengths.

We examine the correlation between the data distribution and the representation distribution. Using Laplace data as input and observing Gaussian representations at the output, we can compute likelihoods for both input and output sets. Comparing these scores reveals strong correlation (Fig. 11), indicating that the distribution is indeed "pushed forward" through the encoder. This correlation remains stable across different augmentation strengths, showing that this "pushforward" behavior is insensitive to the level of augmentation.

