# OpenReview forum: "InfoNCE Induces Gaussian Distribution"
_ICLR.cc/2026/Conference — ICLR 2026 Oral_

### Official Review · Reviewer_yqFV · 2025-10-24

**Soundness:** 3
**Presentation:** 2
**Contribution:** 2
**Rating:** 4
**Confidence:** 4

**Summary:**

This paper investigates the statistical structure of representations learned through contrastive learning with the InfoNCE loss, and demonstrates that such representations closely follow a multivariate Gaussian distribution. The authors provide two theoretical justifications: (1) under alignment and concentration assumptions, high-dimensional embeddings asymptotically approach Gaussianity; and (2) under weaker assumptions, introducing a small regularization term that encourages low feature norms and high entropy leads to similar asymptotic behavior. Theoretical analysis is further supported by experiments on synthetic data, CIFAR-10, and pretrained foundation models, consistently validating the Gaussian property of learned representations.

**Strengths:**

1.	Theoretical and empirical results align well. The experiments on synthetic datasets, CIFAR-10, and pretrained models convincingly support the Gaussianity of contrastive representations.
2.	The paper is well-organized, with a clear logical flow that makes both the theoretical derivations and empirical validations easy to follow.
3.	The work introduces a novel and meaningful perspective for understanding the contrastive learning objective, potentially offering a new theoretical framework for future studies in this area.

**Weaknesses:**

1.	While the theoretical findings are solid, the paper could benefit from more discussion on how the Gaussianity of learned representations may translate into practical insights or applications. In the introduction, the authors claim that understanding representation distributions is not only of theoretical insights. However, the main body does not explore how this insight could improve contrastive learning methods or be leveraged in practice  This is my main concern.
2.	The claim that contrastive representations follow a Gaussian distribution offers a novel theoretical view. However, it would strengthen the paper to compare this finding to prior work that also studies the representations of contrastive embeddings—such as [1] and [2], which show that contrastive features tend to form well-separated clusters aligned with supervised labels. These perspectives seem complementary: clustering and Gaussianity both describe structural regularities in learned features. Intuitively, it seems that their results could provide more guidance for our application as contrastive learning (as classification is one of the most important application and metric of learned representations). I wonder what are the advantages of the theoretical analysis in this paper. A clearer comparison could help clarify the added value or distinct implications of the Gaussian model over existing analyses.
3.	The paper focuses on the unnormalized representation space, which is an interesting and often underexplored aspect. Most contrastive frameworks compute losses in the normalized space, while fine-tuning typically uses unnormalized embeddings. It would be valuable if the authors could discuss whether their theoretical analysis provides any insight into this transition—e.g., why Gaussianity might hold or break across normalized and unnormalized spaces, and what this implies for the design of pretraining-finetuning pipelines.
4.	The paper progressively relaxes its assumptions. Nevertheless, it remains unclear how strong or realistic these assumptions are in practice. It would be helpful for the authors to clarify the degree of relaxation in each step and to discuss how far the theoretical conditions deviate from those typically observed in real contrastive learning scenarios. More empirical verifications or discussions of assumption validity could make the theoretical results more convincing and grounded.

[1] Saunshi, Nikunj, et al. "A theoretical analysis of contrastive unsupervised representation learning." International conference on machine learning. PMLR, 2019.

[2]  HaoChen J Z, Wei C, Gaidon A, et al. Provable guarantees for self-supervised deep learning with spectral contrastive loss[J]. NeurIPS, 2021.

**Questions:**

See Weaknesses.

---

> ### Author Response · Authors · 2025-11-20
> **Response (1)**
>
> We thank the reviewer for the feedback. We have revised the paper (revisions appear in yellow) according to the weaknesses pointed out by the reviewer. Below we address each weakness seperately.
>
> **Weakness:**
> While the theoretical findings are solid, the paper could benefit from more discussion on how the Gaussianity of learned representations may translate into practical insights or applications. In the introduction, the authors claim that understanding representation distributions is not only of theoretical insights. However, the main body does not explore how this insight could improve contrastive learning methods or be leveraged in practice This is my main concern.
>
> **Response:**
> We agree that connecting Gaussianity to practical benefits is important, and we understand the reviewer’s concern that the applicative side was under-emphasized in the original submission. In the revised version, we add a discussion in the Introduction (lines 35-48) to more explicitly motivate our work and highlight how our results support practical methods that already exploit Gaussianity in contrastive representations.
>
> First, recent work by Eftekhari \& Papyan [1] shows that explicitly Gaussianizing representations can improve downstream image classification performance, providing direct empirical evidence that “more Gaussian” features can be beneficial in practice. Our analysis offers a principled explanation for why such Gaussianization is a natural feature for contrastive representations.
>
> Second, a growing body of work already assumes (often implicitly) that CLIP-like contrastive representations are approximately Gaussian, and builds practical methods on top of this assumption. Examples include:
> - *Uncertainty estimation and Bayesian modeling:* Laplace posteriors over CLIP heads and Bayesian adapters with Gaussian variational parameters[2, 3].
> - *Probabilistic embeddings and prompt learning:* modeling CLIP features or prompts as Gaussians in latent space[4, 5].
> - *Classification and class-incremental learning:* methods that treat per-class CLIP features as Gaussian for replay, generative classifiers, or prototype-based decision rules[6].
> - *Test-time adaptation and calibration:** approaches that use Gaussian priors over class prototypes or feature clusters[7].
> - *Segmentation, detection, and dense prediction:* works that interpret modality- or prompt-specific embeddings as Gaussian latent variables [8, 9].
>
> Third, a Gaussian characterization has direct algorithmic and computational advantages: it makes many otherwise intractable geometric quantities analytically tractable. Under a Gaussian model, entropy, likelihoods, Mahalanobis distances, and KL divergences all admit closed-form expressions, which can be used for OOD detection, scoring, confidence calibration, and density-based diagnostics. For example, [10] explicitly computes image likelihoods from a Gaussian approximation in the embedding space.
>
> Our results provide theoretical support for these practices by explaining why Gaussian modeling of contrastive representations is structurally aligned with the InfoNCE loss, rather than merely heuristic, and they naturally motivate future contrastive objectives or regularizers that explicitly encourage isotropy and Gaussianity (as in [1, 3, 10]) to further improve representation quality.
>
> [1] D. Eftekhari, and V. Papyan. On the Importance of Gaussianizing Representations. ICML 2025.
>
> [2] Anton Baumann et al., Post-hoc Probabilistic Vision–Language Models, arXiv:2412.06014 (2024)
>
> [3] Pablo Morales-Álvarez et al., BayesAdapter: enhanced uncertainty estimation in CLIP few-shot adaptation, arXiv:2412.09718.
>
> [4] Aishwarya Venkataramanan et al., Probabilistic Embeddings for Frozen Vision-Language Models: Uncertainty Quantification with Gaussian Process Latent Variable Models, UAI 2025.
>
> [5] Yuning Lu et al., Prompt distribution learning, Proceedings of the IEEE/CVF conference on computer vision and pattern recognition 2022.
>
> [6] Zitong Huang et al., Learning prompt with distribution-based feature replay for few-shot class-incremental learning, arXiv:2401.01598.
>
> [7] L. Zhou et al., Bayesian test-time adaptation for vision-language models, Proceedings of the Computer Vision and Pattern Recognition Conference 2025.
>
> [8] C. Huang et al.,Multimodal representation distribution learning for medical image segmentation, Proceedings of the Thirty-Third International Joint Conference on Artificial Intelligence 2024.
>
> [9] M. Jia et al., Orchestrating the Symphony of Prompt Distribution Learning for Human-Object Interaction Detection, Proceedings of the AAAI Conference on Artificial Intelligence 2025.
>
> [10] R. Betser, et al., Whitened CLIP as a Likelihood Surrogate of Images and Captions. ICML 2025.

---

> ### Author Response · Authors · 2025-11-20
> **Response (2)**
>
> **Weakness:** The claim that contrastive representations follow a Gaussian distribution offers a novel theoretical view. However, it would strengthen the paper to compare this finding to prior work that also studies the representations of contrastive embeddings, such as [1] and [2], which show that contrastive features tend to form well-separated clusters aligned with supervised labels. These perspectives seem complementary: clustering and Gaussianity both describe structural regularities in learned features. Intuitively, it seems that their results could provide more guidance for our application as contrastive learning (as classification is one of the most important application and metric of learned representations). I wonder what are the advantages of the theoretical analysis in this paper. A clearer comparison could help clarify the added value or distinct implications of the Gaussian model over existing analyses.
>
> **Response:**
> Prior work such as Saunshi et al. and HaoChen et al. analyzes contrastive learning from a task-driven perspective: they study when contrastive objectives yield representations that are linearly separable or cluster according to supervised labels. These results characterize the class-conditional geometry of the learned features, whereas our work characterizes the marginal distribution of the representation itself. The two perspectives are fundamentally complementary: class structure describes how different semantic modes are organized, while our Gaussianity result describes the global high-dimensional law that individual coordinates and random projections converge to.
>
> Importantly, these viewpoints are not in tension. In high dimensions, Gaussian representations naturally support class-conditional clustering and even linear separability. Thus, class separation is fully compatible with, and often expected under, an approximately Gaussian marginal law. Clustering captures semantic organization; Gaussianity captures the underlying ambient geometry.
>
> We now explicitly incorporate both works into the Related Work section (lines 105-111) and clarify that they address different but compatible aspects of contrastive representations. Clustering behavior and label alignment do not determine the marginal probabilistic law of the embeddings; class-conditional clusters can reside inside many possible distributions. Our analysis instead targets this marginal, model-agnostic statistical structure that contrastive objectives impose regardless of labels or downstream tasks. This is precisely the structure exploited by Gaussian-based scoring and calibration methods (e.g., Mahalanobis OOD, Gaussian classifiers, likelihood-style diagnostics).
>
> **Weakness:** The paper focuses on the unnormalized representation space, which is an interesting and often underexplored aspect. Most contrastive frameworks compute losses in the normalized space, while fine-tuning typically uses unnormalized embeddings. It would be valuable if the authors could discuss whether their theoretical analysis provides any insight into this transition, e.g., why Gaussianity might hold or break across normalized and unnormalized spaces, and what this implies for the design of pretraining-finetuning pipelines.
>
> **Response:**
> We agree this normalized-unnormalized transition is important in practice, and our theory does provide insights into it. Our analysis explicitly treats both normalized and unnormalized features. Both of our theoretical routes, the alignment plateau argument (Sec. 4.1) and the regularized objective (Sec. 4.2), yield Gaussianity in each space. Corollary 1 shows that normalized features on the sphere have Gaussian
> k-dimensional projections, while Proposition 2 extends this to the unnormalized representations under thin-shell concentration. Conversely, under the regularized route the unnormalized features are approximately isotropic (Proposition 3), and normalization simply projects this distribution onto the sphere, where the spherical CLT again gives Gaussian marginals. Thus, in both analyses the same statistical geometry emerges features inherit Gaussian projections.
>
> Empirically, this theoretical picture is reflected in our experiments. Across all datasets (synthetic, CIFAR-10, and pretrained CLIP/DINO), we observe that both normalized and unnormalized embeddings are strongly Gaussian, precisely as the predicted theory. We make this connection explicit in the revision (lines 210-213, 405-411).
> Finally, this has direct implications for practice: contrastive objectives benefit from normalized embeddings during pretraining (stability, uniformity), while downstream tasks typically operate on unnormalized embeddings (some downstream tasks also operate on the normalized embeddings). Both spaces exhibit closely related Gaussian behavior, which supports common choices such as linear probing, Mahalanobis scoring and likelihood-style diagnostics.

---

> ### Author Response · Authors · 2025-11-20
> **Response (3)**
>
> **Weakness:**
> The paper progressively relaxes its assumptions. Nevertheless, it remains unclear how strong or realistic these assumptions are in practice. It would be helpful for the authors to clarify the degree of relaxation in each step and to discuss how far the theoretical conditions deviate from those typically observed in real contrastive learning scenarios. More empirical verifications or discussions of assumption validity could make the theoretical results more convincing and grounded.
>
> **Response:**
> We agree that the realism of our assumptions is important, and we now clarify both the ``strength’’ of each assumption and its empirical status. Conceptually, our analysis follows two routes: (i) the plateau route (Sec. 4.1), which assumes an alignment plateau and thin-shell concentration, and (ii) the regularized route (Sec. 4.2), which replaces exact plateau behavior with a weaker bounded-alignment condition and a small regularizer. In the revision we explicitly summarize this hierarchy at the beginning of Sec. 4 (lines 210-222), emphasizing that the plateau assumptions are sufficient but not necessary for Gaussianity, whereas the regularized route requires strictly weaker conditions.
>
> In the first route, we assume that alignment converges before uniformity (Assumption 1), which is precisely the behavior observed in Fig. 2, and that unnormalized embeddings exhibit norm concentration, as shown in multiple experiments (Figs. 3,4,6). We acknowledge that the alignment plateau may not always hold in practice; this is why the second, regularized route is introduced. There we only assume (Assumption 3) that there exists a representation where when directions are uniform the alignment term attains its upper bound (Equation 8, Proposition 1). This is a purely theoretical realizability assumption about the existence of such a representation, not a statement about a particular training trajectory, and under it no explicit regularization is required (the regularization coefficient can be set to zero). We clarify these relationships in the revised version so that the role and realism of each assumption are transparent.
> Together, these clarifications make the role, strength, and practical relevance of each assumption fully transparent in the revised manuscript.
>
> **Final note:**
> We thank the reviewer for raising important points and hope that our answers directly address the reviewer’s concerns. We feel the revised manuscript has improved thanks to the reviewers comments and we would appreciate if the reviewer could re-evaluate given these clarifications and enhancements.

---

### Official Review · Reviewer_vUPT · 2025-10-29

**Soundness:** 2
**Presentation:** 2
**Contribution:** 1
**Rating:** 2
**Confidence:** 4

**Summary:**

The authors theoretically and empirically analyze representations learned by the InfoNCE contrastive loss, showing they become approximately Gaussian in high dimensions. They present two approaches to demonstrate their claim. First, under two assumptions: an “alignment plateau” assumption and a "thin-shell" assumption, the population-level InfoNCE solution is isotropic, so normalized features are uniform on the sphere and any fixed-$k$ projection is asymptotically Gaussian as dimension $d\to\infty$. Second, even under weaker conditions, adding a vanishing $\ell_2$-norm/entropy regularizer biases the solution toward isotropy, yielding the same asymptotic Gaussian law. Empirically, the authors validate these claims on synthetic data, CIFAR-10, and pretrained models (e.g. CLIP, DINO). They measure thin-shell concentration of feature norms and apply normality tests (Anderson–Darling and D’Agostino–Pearson) to demonstrate that InfoNCE-trained features indeed exhibit Gaussian statistics in practice. The paper argues this formalizes prior observations of “near-Gaussian” features in contrastive learning and justifies Gaussian modeling for downstream tasks (e.g. OOD detection).

**Strengths:**

- Theoretical rigor: The paper provides a novel and rigorous analysis using high-dimensional probability and spherical CLT tools (e.g. Hirschfeld–Gebelein–Rényi maximal correlation bound, polar KL decomposition, Maxwell–Poincaré CLT). It derives precise conditions (bounded alignment, uniformity on sphere) under which InfoNCE yields isotropic, Gaussian outputs.

- Comprehensive empirical validation: The authors evaluate across diverse settings: a synthetic Laplace dataset, CIFAR-10 with a small encoder, and large pretrained models (supervised ResNet/DenseNet vs self-supervised CLIP/DINO). They consistently observe thin-shell norm concentration and high normality p‑values for InfoNCE-trained features (vs non-Gaussian behavior for supervised models).

- Clarity and organization: The paper is well-structured. Key assumptions and results are clearly stated (e.g. via bullet-point summary of contributions on p.2). The exposition links the theory to concrete phenomena (alignment saturation, feature variance) and shows illustrative figures (e.g. Figure 3) that make the findings intuitive. Definitions and proofs are thorough and mostly easy to follow.

**Weaknesses:**

- Strong asymptotic assumptions: The theory relies on high-dimensional limits and idealized assumptions (infinite negatives, alignment plateau, perfect norm concentration). These may not hold in all practical cases. The authors acknowledge this, noting the results are asymptotic and “alignment plateau and thin-shell concentration… are not guaranteed to hold universally”. It remains unclear how well the Gaussian approximation holds for moderate $d$ or when these assumptions fail.

- Dependence on regularizer: One route to Gaussianity uses an added regularizer (low-norm, high-entropy) that vanishes asymptotically. It’s not fully clear how sensitive the results are to this term in finite dimensions. Practical guidelines for choosing (or removing) this regularizer without harming performance are not discussed. Besides, it would have been interesting to study the impact of this regularizer on downstream classification performances on one or more different datasets (CIFAR10, STL10, IMAGENET100, IMAGENET1K).

- Scope of experiments: Although varied, the experiments mostly focus on vision models and one dataset (MS-COCO) for pretrained features. It would strengthen the paper to test more domains (e.g. other modalities or tasks or non-natural images) to see if Gaussianity is universal for InfoNCE. Additionally, experiments on how violations of assumptions affect outcomes (e.g. weak augmentations) are not shown.

- Overall, the Gaussianity of the latent space is not so surprising, it has been already shown in Wang and Isola, 2021 that the representations would converge toward a uniform hyperspherical distribution. And it is commonly known that there is a close connection between uniform hyperspherical distribution (von-Mises Fisher with concentration equals to 0) and high-dimensional Gaussian distributions known as the soap-bubble-effect as stated by Vershynin, 2018; Lecture Notes on High-Dimensional Data, Sven-Ake Wegner, https://arxiv.org/pdf/2101.05841; Hyperspherical Variational Auto-Encoders, https://auai.org/uai2018/proceedings/papers/309.pdf Davidson et al. 2018; Farquhar et al, 2021 Radial Bayesian Neural Networks: Beyond Discrete Support In Large-Scale Bayesian Deep Learning, https://arxiv.org/abs/1907.00865. The paper would benefit discussing these results and these papers in order to clearly state their contribution.

- It is unclear why CLIP and DINO (what about v2 or v3 ?) exhibits Gaussian distribution properties even though they were not exactly trained like SimCLR: CLIP positive-pair probably implies different alignment propoerties than SimCLR, and DINO's loss is not exactly the same as SimCLR, the paper would benefit from a better discussion about other self-supervised methods.

**Questions:**

- In practice, how does one verify the “alignment plateau” condition during training? Can alignment be measured online to check this assumption? Is the plateau different given different images ? Does that mean something about the quantity of information carried by the image ?

- The vanishing regularizer is claimed to ensure isotropy at finite $d$. How should its strength be chosen in practice? Did you experiment with non-vanishing regularizers, and if so, how did they affect downstream performance?

- The work emphasizes normalized (unit) features. Does the Gaussianity hold equally for unnormalized (pre-norm) embeddings? Fig.3 and authors remarks suggests yes, but clarity on this point would be good. It is unclear why would upstream representations (like the one used for downstream tasks, typically before the projection MLP 2 layers) would exhibit a multi-dimensional Gaussian distribution as well.

- Do these results extend to other contrastive losses ? Empirically, on CLIP and DINO it appears to, but what is the theoretical argument to support this behaviour ? Is the Gaussian outcome specific to InfoNCE batch training?

---

> ### Author Response · Authors · 2025-11-20
> **Response (1)**
>
> We thank the reviewer for the detailed feedback. We have read it carefully and revised the paper (revisions marked in yellow) appropriately, adding clarifications and new experiments. Below we address each weakness and question.
>
> **Weakness:**
> Strong asymptotic assumptions...
>
> **Response:**
> The theory necessarily relies on high-dimensional and asymptotic assumptions, as do most analytical treatments of contrastive learning and self-supervised representation learning. We view these assumptions not as limitations but as the minimal structure required to make the problem mathematically tractable. Importantly, even though the results are asymptotic, the assumed behaviors (alignment plateau, thin-shell concentration) and the predicted result (Gaussianity of representations) are all properties that can be empirically tested.
>
> To this end, we complement the theory with extensive empirical validation. Using standard normality diagnostics (AD and DP tests) and norm-based concentration measures (CV of representation norms), we evaluate how closely real representations adhere to the predicted Gaussian behavior. These tests are applied both to real contrastive encoders and to controlled synthetic settings, allowing us to quantify the degree to which finite-dimensional models approximate the theoretical limit. We added a new experiment, demonstrating that on the same data (CIFAR-10) and the same model (ResNet-18), supervised training does not result with Gaussian embeddings and contrastive training does (Table. 1, and lines 479-485). Overall we examined synthetic data with a liner classifier, CIFAR-10 with a small network (2 layered MLP), CIFAR-10 with ResNet-18 and generalized to foundation self-supervised models (CLIP, DINO).  Across all cases, we observe clear trends toward the predicted asymptotic laws.
>
> We therefore see the asymptotic framework as a principled starting point: it offers a rigorous characterization of the marginal distributional structure induced by the contrastive objective, and it yields concrete predictions that can be assessed empirically. While no finite model can satisfy the asymptotic assumptions perfectly, our experiments show that the Gaussian approximation remains accurate and informative well before the infinite-dimensional limit. We have clarified this point in the revised Experiments and Conclusions sections (Sec. 5, lines 405-411, Sec. 6, lines 520-524, 527-529).
>
>
> **Weakness:**
> Dependence on regularizer...
>
> **Question:**
> The vanishing regularizer is claimed to ensure isotropy at finite
> . How should its strength be chosen in practice? Did you experiment with non-vanishing regularizers, and if so, how did they affect downstream performance?
>
> **Response:**
> We thank the reviewer for raising this point regarding the role of the regularizer. In our theoretical development, the regularizer serves only as a mathematical device and, under Assumption 3, vanishes in the asymptotic regime. In practice, contrastive models already employ implicit or explicit norm controls (e.g., weight decay, normalization layers), and our theory clarifies how these common mechanisms interact with the geometry of the InfoNCE objective.
>
> To evaluate finite-dimensional sensitivity, we added a new experiment on CIFAR-10 in which we vary the strength of weight decay (none, standard, and strong). Across all regimes, the resulting embeddings exhibit clear convergence toward Gaussian behavior according to our AD, DP, and CV diagnostics. This indicates that the emergence of approximate Gaussianity is robust and does not rely on fine-tuned regularization (Table 1). However, we do observe that removing weight decay leads to higher norm values, suggesting that some degree of regularization is beneficial in practice.
> Because the asymptotic analysis shows that the regularizer disappears in the limit, and our empirical results confirm that Gaussianity emerges regardless of weight decay, we conclude that Gaussianity is primarily a property of the contrastive objective rather than of the specific regularizer. We have incorporated these findings and clarifications into the revised Experiments section (lines 463-485).
>
> Finally, we note that exploring explicit regularizers (such as those used in our derivation, including an entropy-based term) could be an intriguing direction for future work. Light regularization may promote Gaussianity earlier in training without harming downstream performance, but fully understanding this remains open.

---

> ### Author Response · Authors · 2025-11-20
> **Response (2)**
>
> **Weakness:**
> Scope of experiments...
>
> **Response:**
> We thank the reviewer for this suggestion. Extending the empirical scope is indeed valuable, and we have revised the paper to make this clearer. In the text and audio domains, contrastive learning is typically implemented in settings that differ from the visual case: the notion of augmentations and the construction of positive/negative sets are defined quite differently.
> Nonetheless, our work already includes several experiments beyond a single dataset or model class. In addition to MS-COCO features, we evaluate on synthetic distributions (Laplace and newly added Gaussian mixtures; see Table 1, and Figures 2,3) and on CIFAR-10 using supervised and contrastive trainings )Table 1, and Fig. 4).
>
> To further address the reviewer’s concern, we added a new experiment in which we train ResNet-18 in a SimCLR-style setting across a range of augmentation strengths. We observe that stronger augmentations lead to lower alignment values, while weaker augmentations yield higher alignment. This is inline with our bounded alignment theory (Proposition 1, Equation 7). However, in all cases, the resulting representations remain approximately Gaussian according to our diagnostics (Table 1). This directly examines how different augmentation types affect the outcome.
>
> We also expanded our evaluation to a non-natural image domain by encoding painting and sketch images from ImageNet-R (lines 511-514) using CLIP encoder. These images differ substantially from standard natural images in structure and texture, yet we again observe approximate Gaussianity of the embeddings (Table 1). This provides additional evidence that the phenomenon is not limited to a single dataset or modality of visual appearance.
>
> Together, this set of experiments shows that approximate Gaussianity is robust across datasets, augmentation regimes, and even non-natural image domains, further supporting the universality of the effect under InfoNCE-style representations.
>
>
> **Weakness:**
> Overall, the Gaussianity of the latent space is not...
>
> **Response:**
> We thank the reviewer for highlighting the connection to uniform hyperspherical distributions and the “soap-bubble” intuition from high-dimensional probability. Indeed, Wang and Isola, 2021 note that normalized contrastive embeddings tend toward a uniform distribution on the sphere, however they only prove this for part of the InfoNCE loss (the uniform part in our Equation 5). Other works show that uniform hyperspherical laws share certain geometric properties with high-dimensional Gaussians. We fully agree that this intuition makes Gaussianity plausible.
>
> However, to the best of our knowledge, none of these works provides a theoretical derivation showing that the representations produced by the InfoNCE objective (with and without normalization) converge to a Gaussian marginal distribution. Our contribution is to formalize and derive these properties under our explicit assumptions. We believe establishing this distribution-level characterization is valuable precisely because it makes explicit the assumptions under which Gaussianity emerges, enabling future work to relax or refine them. We have updated the Related Work section (lines 99-112) to discuss these prior results (partially already cited in the original version) and to clearly distinguish their geometric intuition from our distributional analysis.
>
> **Question:**
> In practice, how does one verify ...
>
> **Response:**
> The realism of our assumptions is indeed important. In practice, alignment can be monitored online by computing the mean cosine similarity between positive pairs during training. As shown in Fig. 2, alignment typically converges much earlier than uniformity. This empirical behavior motivates our assumption that alignment reaches a plateau prior to the uniformity dynamics dominating. We also verify norm concentration of unnormalized embeddings across multiple experiments (Table 1, Figures 3,4,6), providing additional evidence that the theoretical regime we analyze is relevant to real models.
> We emphasize that the alignment plateau may not hold universally in all settings. This is precisely why we introduce a second, regularized derivation: under Assumption 3, we only require the existence of a representation whose directions are uniform and whose alignment attains its upper bound (Proposition 1, Equation 7), without assuming that the actual training dynamics achieve perfect convergence.
>
> Finally, the alignment plateau is directly connected to the alignment bound established in Equation 7, which depends on the augmentation channel. Stronger augmentations lower the achievable alignment bound (verified in the new SimCLR setting experiment, lines 479-485), although this does not necessarily imply that the plateau is reached earlier (or at all). We hope this clarifies the relationship between augmentation strength, alignment dynamics, and the assumptions used in our theoretical analysis.

---

> ### Author Response · Authors · 2025-11-20
> **Response (3)**
>
> **Weakness:**
> It is unclear why CLIP and DINO ...
>
> **Question:**
> Do these results extend to other contrastive losses?...
>
> **Response:**
> We appreciate this comment and have updated the experiment section accordingly (lines 478-486, 499-514).
> First, we now include a standard SimCLR experiment on CIFAR-10 in the main paper. SimCLR matches our theoretical setting most closely (single encoder, InfoNCE), so it is a natural testbed. In the revision we show that SimCLR exhibits precisely the behavior predicted by our theory: strong thin-shell concentration and near-Gaussian diagnostics for normalized and unnormalized features. This directly addresses the concern that the ``ideal'' case was not represented in the main experiments.
>
> DINO and CLIP are included to test how far these geometric conclusions extend beyond the theoretically analyzed single-encoder InfoNCE setting. We start from a synthetic case, move on to real data but a small 2-layer MLP. Then SimCLR with ResNet-18 and lastly we demonstrate on self-supervised foundation models.
> For DINO, our intent was not to claim a full theory for DINO, but to treat it as an out-of-scope yet informative baseline: despite its different loss (teacher-student distillation with centering and sharpening), its embeddings empirically satisfy the same high-dimensional geometry (thin-shell and Gaussian diagnostics). We clarify this in the revision (Sec. 5 lines 504-506) and explicitly position DINO as an empirical indication that the geometric picture may extend beyond pure InfoNCE, motivating future theoretical work on distillation-based SSL. We also note that tests on DINOv2 and DINOv3 show the same Gaussian behavior.
> For CLIP, the setting is indeed more complex: two different encoders, two spaces, and a bi-modal contrastive objective. Our Equation 5 is written for the single-encoder population abstraction, and we now explicitly state that this is the scope of the formal results. In CLIP, however, each modality still participates in an InfoNCE-style loss with a uniformity potential over similarities, and empirically each marginal (image, text) exhibits the same Gaussian structure as in the single-encoder case. In the revision we (i) add a short discussion in Sec. 5 clarifying that CLIP lies partially outside the formal setting (shared encoder, single space), and (ii) reframe our CLIP results as evidence that the same geometric phenomena appear in more realistic multi-modal systems rather than as a direct consequence of the current theorems (lines 499-514).
>
> Overall, SimCLR now serves as the ``theory-matched'' experiment, while CLIP and DINO are clearly presented as generalization and practical relevance checks. We believe this clarifies the scope of the theory, addresses the concern about missing SimCLR, and explains why we still find it valuable to report Gaussianity for DINO and CLIP even though a full theoretical treatment of these architectures is left for future work. At the same time, the consistent Gaussian behavior observed in these broader settings explicitly motivates extending our analysis beyond single-encoder InfoNCE to additional self-supervised settings, including multi-modal objectives, which we now highlight as a concrete direction for future theoretical work.
>
> **Question:**
> The work emphasizes normalized ..
>
> **Response:**
> We agree this normalized-unnormalized transition is important in practice, and our theory does provide insights into it. Our analysis explicitly treats both normalized and unnormalized features. Both of our theoretical routes, the alignment plateau argument (Sec. 4.1) and the regularized objective (Sec. 4.2), yield Gaussianity in each space. Corollary 1 shows that normalized features on the sphere have Gaussian
> k-dimensional projections, while Proposition 2 extends this to the unnormalized representations under thin-shell concentration. Conversely, under the regularized route the unnormalized features are approximately isotropic (Proposition 3), and normalization simply projects this distribution onto the sphere, where the spherical CLT again gives Gaussian marginals. Thus, in both analyses the same statistical geometry emerges features inherit Gaussian projections.
>
> Empirically, this theoretical picture is reflected in our experiments. Across all datasets (synthetic, CIFAR-10, and pretrained CLIP/DINO), we observe that both normalized and unnormalized embeddings are strongly Gaussian, precisely as the predicted theory. We make this connection explicit in the revision (lines 210-213, 405-411).
> Finally, this has direct implications for practice: contrastive objectives benefit from normalized embeddings during pretraining (stability, uniformity), while downstream tasks typically operate on unnormalized embeddings (some downstream tasks also operate on the normalized embeddings). Both spaces exhibit closely related Gaussian behavior, which supports common choices such as linear probing, Mahalanobis scoring and likelihood-style diagnostics.

---

> ### Author Response · Authors · 2025-11-20
> **Response (4)**
>
> **Final note:**
> We thank the reviewer again for the thorough and constructive feedback, which has helped us improve the paper. We believe the revised manuscript is substantially strengthened, with clearer exposition and additional experiments that directly address the concerns raised. We would welcome any further feedback, and, if the reviewer judges that the revised version better meets their expectations, we would kindly ask them to reconsider the initial rating.

---

### Official Review · Reviewer_BFsh · 2025-10-31

**Soundness:** 4
**Presentation:** 3
**Contribution:** 3
**Rating:** 8
**Confidence:** 4

**Summary:**

This paper studies the distribution of the optimal representation obtained by optimizing the InfoNCE objective. It demonstrates, under alignment and concentration assumptions, that finite projections of the high-dimensional representation are Gaussian. The authors then weaken their alignment and concentration assumption to obtain a similar result. Finally, they give empirical evidence to confirm they claim on widely used pre-trained contrastive models (CLIP) and non-contrastive (DINO) models.

**Strengths:**

-	The mathematical analysis is rigorous, the first claims (Corollary 1 and Proposition 2) are strong (even if they are directly obtained from two well-known results) and the section 4.2 is very technical but sounded.
-	The empirical evidence given for real-world contrastive-based models gives credit to the theoretical analysis. There are not many works testing the Gaussian assumption on the representations of foundation models while it is often assumed for downstream applications (e.g. OOD).
-	The exposition is clear, concise, and easy to follow (excepted maybe for section 4.2 which requires special attention for the numerous notations…).

**Weaknesses:**

-	Regarding Assumption 3, the authors mentioned it is weaker than previous Assumption 1. It is still not clear to me the realizability of such assumption and why it would hold in practice. Assumption 1 may be easier to verify empirically (as the authors already mentioned) and it seems more intuitive.
-	In the experiments, you consider DINO, a non-contrastive approach that does not introduce any uniformity term in its loss (which is key in your analysis since you always assume the alignment to be controlled by a constant). It is interesting to note that DINO representation seems to be Gaussian as well, but you do not justify it theoretically. Do you have any theoretical analysis linking the knowledge distillation technique in DINO with the uniformity loss optimized by InfoNCE?
-	In your experiments, you also consider CLIP, which is a constative approach, but it does not fit well in your theoretical analysis. Indeed, the vision and text encoders are different (they do not share the same weights, and they are not defined in the same space) and the marginal distribution for each modality is different. Therefore, your uniformity term in eq. 5 seems to not fit this case. It is again interesting to note that image and text representations are Gaussian but you do not justify it theoretically.
-	Finally, it is funny to observe that the only foundation model that fits perfectly your analysis (SimCLR) is not present in your experiments!

**Questions:**

Most of my questions are described in the weaknesses. I believe answering them will improve the quality of the manuscript, in particular regarding the match to the current practice in contrastive learning.

---

> ### Author Response · Authors · 2025-11-20
> **Response**
>
> We thank the reviewer for the constractive feedback. We have revised the paper (additions and edits in yellow) according to the raised points. Below we address each weakness.
>
> **Weakness:**
> Regarding Assumption 3, the authors mentioned it is weaker than previous Assumption 1...
>
> **Response:**
> We appreciate the request for clarification regarding Assumption 3 and its realizability. Assumption 3 is indeed weaker than Assumption 1, but its role in the analysis is different. Assumption 1 (alignment plateau) is an empirically grounded training-time assumption: it describes a behavior that can be directly observed and verified in practice (and which we document in Fig. 2). By contrast, Assumption 3 is not intended as a description of the training trajectory. It is a theoretical realizability assumption: it requires only that a representation exists whose alignment attains the upper bound and whose directions are uniform. The assumption therefore does not need to hold in the optimization path, only that the objective admits such a solution.
> Because Assumption 3 asserts existence rather than training behavior, it is strictly weaker than Assumption 1. It also does not require any additional structure: the regularization coefficient in Sec. 4.2 can be taken arbitrarily small or even zero whenever this assumption holds, and when it does not, Equation 20 quantifies the strength of regularization needed to enforce it.
>
> **Weakness:**
> In the experiments, you consider DINO...
> **Weakness:**
> In your experiments, you also consider CLIP...
> **Weakness:**
> Finally, it is funny to observe that the only foundation model that fits perfectly your analysis (SimCLR) is not present in your experiments!
>
> **Response:**
> We appreciate these comments and have updated the experiments and discussion accordingly.
>
> First, we now include a standard SimCLR experiment on CIFAR-10 in the main paper. SimCLR matches our theoretical setting most closely (single encoder, InfoNCE), so it is a natural testbed. In the revision we show that SimCLR exhibits precisely the behavior predicted by our theory: strong thin-shell concentration and near-Gaussian diagnostics for normalized and unnormalized features. This directly addresses the concern that the ``ideal'' case was not represented in the main experiments.
>
> DINO and CLIP are included to test how far these geometric conclusions extend beyond the theoretically analyzed single-encoder InfoNCE setting. We start from a synthetic case, move on to real data but a small 2-layer MLP. Then SimCLR with ResNet-18 and lastly we demonstrate on self-supervised foundation models.
>
> For DINO, our intent was not to claim a full theory for DINO, but to treat it as an out-of-scope yet informative baseline: despite its different loss (self- distillation), its embeddings empirically satisfy the same high-dimensional geometry (thin-shell and Gaussian diagnostics). We clarify this in the revision (Sec. 5 lines 499-514) and explicitly position DINO as an empirical indication that the geometric picture may extend beyond pure InfoNCE, motivating future theoretical work on distillation-based SSL.
>
> For CLIP, the setting is indeed more complex: two different encoders, two spaces, and a bi-modal contrastive objective. Our Equation 5 is written for the single-encoder population abstraction, and we now explicitly state that this is the scope of the formal results. In CLIP, however, each modality still participates in an InfoNCE-style loss with a uniformity potential over similarities, and empirically each marginal (image, text) exhibits the same Gaussian structure as in the single-encoder case. In the revision we (i) add a short discussion in Sec. 5 clarifying that CLIP lies partially outside the formal setting (shared encoder, single space), and (ii) reframe our CLIP results as evidence that the same geometric phenomena appear in more realistic multi-modal systems rather than as a direct consequence of the current theorems (lines 499-514).
>
> Overall, SimCLR now serves as the ``theory-matched'' experiment, while CLIP and DINO are clearly presented as generalization and practical relevance checks. We believe this clarifies the scope of the theory, addresses the concern about missing SimCLR, and explains why we still find it valuable to report Gaussianity for DINO and CLIP even though a full theoretical treatment of these architectures is left for future work. At the same time, the consistent Gaussian behavior observed in these broader settings explicitly motivates extending our analysis beyond single-encoder InfoNCE to additional self-supervised settings, including multi-modal objectives, which we now highlight as a concrete direction for future theoretical work.
>
> **Final note:**
> We thank the reviewer for the important comments, which helped us improve the clarity of the paper, especially in the experiments section. We appreciate the thoughtful feedback and believe the revised version is substantially strengthened as a result.

---

> > ### Comment · Reviewer_BFsh · 2025-11-28
> >
> > I would like to thank the authors for including my remarks and adding the new experiments to the manuscript. I believe it strengthens the contributions of this work. I stick to my rating because I consider that this paper is shedding light on an important property of the InfoNCE objective. To me, it also completely aligns with the recent empirical observations made by LeJEPA.

---

### Official Review · Reviewer_j58r · 2025-11-01

**Soundness:** 2
**Presentation:** 2
**Contribution:** 2
**Rating:** 2
**Confidence:** 3

**Summary:**

The paper presents theory that representations obtained with contrastive learning yield "Gaussian representations". The authors theoretially investigate this claim for normalized and unnormalized contrastive learning techniques and present a small scale theoretical study across popular contrastive learning models in computer vision.

**Strengths:**

- The authors connect new theory on properties of contrastive representations with empirical experiments covering real world models
- The paper is dense with theoretical results, but still fairly well structured and easy to follow
- The authors release code for reproducibility

**Weaknesses:**

1. It seems like the authors missed existing identifiability literature for contrastive learning, e.g. https://arxiv.org/abs/1605.06336, https://proceedings.mlr.press/v54/hyvarinen17a/hyvarinen17a.pdf, https://arxiv.org/abs/1805.08651, https://proceedings.mlr.press/v139/zimmermann21a.html, https://arxiv.org/abs/2007.00810, https://arxiv.org/pdf/2410.21869. It would be good to discuss how the presented theory (which uses a different set of tools) relates to this work; the presented work does not talk about the data generating process, but I wonder if the results are fundamentally connected to identifiability; Gaussian representations (or their projections to the sphere) is what we would expect with the considered form of the InfoNCE loss (in particular cf. https://proceedings.mlr.press/v139/zimmermann21a.html) for this.
2. "In particular, the raw unnormalized representations have received little theoretical attention": This statement seems ungrounded, indeed a lot of identifiability theory also studies the unnormalized case (see above).
3. The experiments are not sufficiently rigorous to support the theory. It is unclear to me what the null hypothesis is, and how the authors test for Gaussianity. It would be quite helpful if a study with simulated data would be conducted that precisely illustrates the theoretical claims.

**Questions:**

1. I might have a fundamental misunderstanding here, but the sentence "Overall, we provide the first principled explanation for Gaussianity in contrastive representations" seems off. Can you comment on how your theory relates to identifiability theory for contrastive learning, which
2. What is the null hypothesis against the Gaussian distribution that is being observed, what would be the alternative? Will the Gaussian distribution also appear when the underlying data-generating process is non-gaussian distributed, or cannot me mapped into a feature space which is Gaussian-distributed?
3 What is a "Gaussian projection"? (L 191) I assume this is a low-D projection of the full feature space on a 1D axis, where Gaussianity is tested? Doesn't the Gaussianity of a projection then trivially follow from the law of large numbers? If not, could you highlight why this finding is significant?
4. What is the expectation in Eq. 8 over?
5. Consider a mixture of Gaussian distribution, or a vMF mixture where each component represents one class; can you comment if the marginal distribution expected through contrastive learning for this case would be Gaussian? Previous work seems to suggest that a mixture of Gaussians would be recovered, and this seems to conflict with the presented theory here; could you comment?

---

> ### Author Response · Authors · 2025-11-20
> **Response (1)**
>
> We thank the reviewer for the important comments and questions. We have revised the manuscript (revisions marked yellow), added experiments and adjusted statements according to the review. Below we address each weakness and question.
>
> **Weakness:**
> It seems like the authors missed existing identifiability literature for contrastive learning, e.g. ... Gaussian representations (or their projections to the sphere) is what we would expect with the considered form of the InfoNCE loss (in particular cf. https://proceedings.mlr.press/v139/zimmermann21a.html) for this.
>
> **Question:**
> I might have a fundamental misunderstanding here, but the sentence "Overall, we provide the first principled explanation for Gaussianity in contrastive representations" seems off. Can you comment on how your theory relates to identifiability theory for contrastive learning, which"
>
> **Response:**
> We thank the reviewer for highlighting the identifiability literature. We agree that this line of work is relevant background, and we now discuss it explicitly in the revised Related Work (lines 99-111), such that all the provided papers are currently been cited.
>
> *Relation to identifiability theory.*
> Identifiability papers study whether the latent variables of a structured statistical model are uniquely determined by the observed data distribution, up to inherent symmetries. In contrastive-learning identifiability work, the question is: under specific structural assumptions on the data (e.g., latent sources, augmentation channel), does optimizing a given contrastive objective recover those latent variables (up to allowable invariances)? Specifically, these works do not ask what is the resulted representation distribution.
>
> Our question is fundamentally different: *given the InfoNCE loss, what distributional geometry does it induce on the representations at the population optimum?* In particular:
> We do *not* assume any data generating process and we do *not* study recovery of latent factors.
> We analyze the *population InfoNCE functional itself* and characterize the marginal distribution of its minimizers.
>
> Identifiability results do not aim to characterize the marginal law of the learned representations or their asymptotic shape; instead, they focus on which latent variables can be recovered under structural assumptions. Our contribution is complementary: we characterize the distributional geometry induced by InfoNCE at the population optimum and, under our assumptions, show that it leads to isotropic representations with Gaussian projections.
>
> Regarding the remark that "Gaussian representations (or their projections to the sphere) is what we would expect" for InfoNCE, in particular in light of Zimmermann et al. We agree, but although it could be logical or anticipated, we couldn't find any available theoretical derivation for this. We think it is important to theoretically establish this distributional ground, even under certain assumptions, in order to be able to know under which assumptions, the converged distribution is expected to be Gaussian. Hopefully, follow-up works could alleviate some of the assumptions.
> Concretely, The results in Zimmermann et al., and other works are identifiability statements: under a latent-variable model, they show that contrastive learning can recover latent factors (up to symmetries), but they do not characterize the marginal distribution of the learned representations and in particular do not imply that these are (approximately) Gaussian. Even when identifiability holds, recoverability of latent factors is compatible with many different marginal geometries. By contrast, our contribution is to show that, under our assumptions and without specifying any data generating model, the minimizers of the population InfoNCE functional exhibit isotropy and asymptotically Gaussian projections. This Gaussian geometry follows from tools such as HGR maximal correlation and the Maxwell-Poincaré CLT, and is therefore not a direct corollary of existing identifiability results. To isolate our contribution from large datasets typically assumed in identifiability theory, please see Table 1, where a InfoNCE trained network trained in a small data regime (CIFAR-10), converges to Gaussian while supervised based training (with the same architecture and data) does not.
>
> *Clarifying our statement.*
> We agree that the phrasing "first principled explanation" was too strong. In the revision (lines 538-539) we replaced it with:
>
> > “Overall, We provide a principled asymptotic explanation for Gaussianity in contrastive representations”
>
> The revised Related Work now includes a paragraph explicitly connecting to the cited identifiability papers and clarifying that identifiability addresses recoverability, whereas our analysis addresses the distributional limit of representations induced by InfoNCE, thanks again for pointing it out.

---

> > ### Comment · Reviewer_j58r · 2025-11-22
> > **Re: Response (1), identifiability**
> >
> > Dear authors, thanks for the reply. I would like to continue the discussion of the identifiability results, as this sentence:
> >
> > > Our question is fundamentally different: given the InfoNCE loss, what distributional geometry does it induce on the representations at the population optimum? In particular: We do not assume any data generating process and we do not study recovery of latent factors.
> >
> > highlights well what I meant in my original review. Even though the presented theory does *not* make assumptions about the particular data generating process, this does not change the fact that InfoNCE minimization recovers a conditional vMF (on the sphere) or conditional Gaussian (in Euclidean space) if there exists a mapping from data to a representation where the latent pairs can be represented as a conditional vMF or conditional Gaussian.
> >
> > However, I see your point that you are mainly talking about the marginals, while e.g. Zimmermann et al. assume uniformity on the hypersphere/convex bodies for most part of the theory. Empirically, however Table 1 shows the case also for Gaussian marginals, showing that InfoNCE also recovers these; I would expect that there might be a follow-up paper discussing this case from the theoretical perspective as well.
> >
> > What would fully convince me here is a case where you generate data in a way such that mapping to a Gaussian marginal+conditional (Euclidean space) or vMF marginal + conditional (hypersphere) is impossible with an arbitrary bijective mapping; minimize InfoNCE; and show that you still recover a Gaussian distribution, indicating the that InfoNCE is biased to this distribution.
> >
> > Happy to discuss more.
> >
> > ---
> >
> > Edit: Referencing your response above,
> >
> > > Instead, Gaussianity arises via the Maxwell-Poincaré spherical CLT, which states that low-dimensional projections of vectors that are (i) uniform in direction and (ii) have fixed norm values, become asymptotically Gaussian. Our analysis shows that InfoNCE drives representations toward these geometric conditions. This explains why contrastive representations exhibit approximately Gaussian distributions in practice, independently of whether the input distribution is Gaussian.
> >
> > this even better fits with the uniformity assumed in Zimmermann et al. -> under uniform marginal on hypersphere/conditional vMF we expect InfoNCE to recover this distribution. Even without assumptions on the data generating process, if the distribution can be represented like this, it will be recovered by the InfoNCE.

---

> > > ### Author Response · Authors · 2025-11-22
> > >
> > > We thank the reviewer for the continued engagement and clarification; this is highly appreciated.
> > >
> > > To directly address the point that our empirical results may simply reflect recoverability of a latent Gaussian or vMF structure under a suitable bijective map, we designed an experiment where such a mapping is impossible by construction. Specifically, we generate a dataset with fully discrete binary vectors. Because the data takes only a finite set of discrete values, it cannot be mapped to a continuous Gaussian or vMF distribution by any invertible transformation. Therefore, if the learned representations exhibit Gaussian structure, it arises from the optimization dynamics of InfoNCE rather than recovery of a latent Gaussian model.
> > >
> > > **Setup:**
> > > - We generate 4,096 samples, each is a binary vector of length 1,024.
> > > - The positive pair is constructed by independently selecting a small number of zero entries (0.1%), and setting them to ones.
> > > - A linear encoder maps inputs to 256-dimensional embeddings, then we train using InfoNCE for 100 epochs.
> > > - Every 10 epochs we compute normality diagnostics: CV, AD, DP statistics
> > >
> > > **Results:**
> > >
> > > | Epoch | Norm CV | AD mean | AD % pass | DP mean p | DP % pass |
> > > |-------|---------|---------|------------|------------|------------|
> > > | 0     | 0.36    | 1.64    | 0.30       | 0.02       | 0.15       |
> > > | 50    | 0.12    | 0.41    | 0.93       | 0.44       | 0.89       |
> > > | 100   | 0.09    | 0.42    | 0.97       | 0.46       | 0.98       |
> > >
> > > **Interpretation:**
> > >
> > > At the start of training, the representation is clearly non-Gaussian: norms vary significantly and most dimensions fail normality tests. As training proceeds, the representation becomes highly Gaussian-like: norms concentrate, and over 95\% of dimensions pass normality tests.
> > >
> > > This occurs despite the data distribution being discrete and not representable as a continuous Gaussian or vMF distribution under any invertible mapping. These results support the interpretation that Gaussian structure is induced by the InfoNCE objective rather than recovered from an underlying latent Gaussian model.
> > >
> > > We also evaluated this setup across multiple configurations, varying input dimensions, embedding dimensions, sparsity levels (fraction of 1's in each data sample), and augmentation strengths. We observe consistent convergence toward Gaussian-like behavior. If the reviewer agrees that this experiment addresses his concern, we will include it in the paper.
> > >
> > > *Final remark:* our theory and results do not contradict identifiability works. We view the identifiability line of research as complementary, and as motivation for further work clarifying how recoverable latent structure relates to the biases that different objectives induce in the learned representations. In particular, we think that explicitly aligning these two perspectives could be a promising direction for future work.  We are also motivated to better understand the latent structure of real image data (e.g., CIFAR-10, ImageNet-R, MS-COCO used in our experiments) and investigate to what extent it is Gaussian or deviates from such models. We see this paper as a first step toward that goal.
> > >
> > > We thank the reviewer for being open to discussion and for the constructive feedback, which we believe has helped us strengthen and clarify the paper. We would be happy to continue the discussion if additional concerns arise.

---

> ### Author Response · Authors · 2025-11-20
> **Response (2)**
>
> **Weakness:**
> The experiments are not sufficiently rigorous to support the theory...
>
> **Question:**
> What is the null hypothesis against the Gaussian distribution that is being observed...
>
> **Response:**
> The reviewer raises an important point about how we test for Gaussianity and how our empirical diagnostics relate to the theory. Testing high-dimensional Gaussianity is inherently challenging, so we employ multiple complementary diagnostics. Throughout all experiments, we use standard one-dimensional normality tests as finite-sample indicators of convergence toward the asymptotic laws predicted by our analysis. The null hypothesis in each case is that an individual coordinate of the representation is Gaussian; the alternative is that it is non-Gaussian. We use
> (i) the Anderson-Darling (AD) test (we fail to reject when AD < 0.752), and
> (ii) the D’Agostino-Pearson (DP) test (we fail to reject when p > 0.05).
> For each model we report both the average statistic and the percentage of coordinates for which the null is not rejected.
>
> In addition, we report the coefficient of variation (CV, Equation 21) of representation norms, which measures thin-shell concentration. In high dimensions, low CV values is characteristic of Gaussian vectors and is a direct consequence of our theoretical predictions.
>
> Taken together, the combination of per-coordinate normality tests with global norm concentration provides a strong practical diagnostic for approximate Gaussianity. Natural alternatives such as Gaussian mixtures, Student-t distributions, and other smooth non-Gaussian families fail at least one of these tests. While it is theoretically possible to construct highly contrived, non-smooth distributions that pass these diagnostics yet are not Gaussian, such cases are extremely unlikely to arise in neural network-based representations.
> We have revised the Experiments section (lines 412-428) to explicitly state these hypotheses, thresholds, and the rationale for using these metrics.
>
> *Simulated data experiment illustrating the theory.*
> Our synthetic Laplace experiment (Sec. 5, Fig. 3, Table 1) is designed specifically to illustrate the theoretical predictions in a controlled setting. The inputs are i.i.d. Laplace (strongly non-Gaussian), and we train a linear encoder with InfoNCE while increasing dimension and batch size. As predicted, (i) the norms concentrate (CV decreases toward zero), and (ii) the AD/DP statistics are in the Gaussian acceptance region, with all coordinates failing to reject the Gaussian null.
> We further repeat the experiment on a synthetic 25-component Gaussian mixture and observe the same pattern: the learned marginal representations pass the AD/DP normality tests and exhibit strong norm concentration. This shows that Gaussianity emerges even when the data-generating process is either heavy-tailed (Laplace) or multimodal (mixture). We added a new table (Table, 1) summarizing these synthetic results.
>
> *Additional empirical evidence.*
> We include a CIFAR-10 experiment with a two-layer MLP (Sec. 5, Fig. 4), where we observe the same behavior: the representations begin far from Gaussian and move steadily toward the Gaussian regime as training proceeds. We also add a new controlled comparison (lines 479-514, Table 1) showing that *supervised* training of a ResNet-18 model on CIFAR-10 does *not* produce Gaussian coordinates nor thin-shell concentration (CV remains high), whereas InfoNCE-based training (SimCLR protocol) produces highly Gaussian coordinates with concentrated norms. This shows that Gaussianity is not a generic consequence of the data, SGD, neural architectures, or high dimensionality, but is tied specifically to the geometric bias induced by InfoNCE, not to a trivial LLN effect. Note that all of the aforementioned experiments are conducted on small sets, underscoring that the Gaussianity is derived by the InfoNCE loss and not by the rule of large numbers.
>
> *Clarifying "Gaussian projections."*
> A Gaussian projection refers to a fixed low-dimensional linear projection of the representation. This does not follow from the Law of Large Numbers (LLN): applies to averages of (approximately) independent variables, which is not the setting here. Instead, Gaussianity arises via the Maxwell-Poincaré spherical CLT, which states that low-dimensional projections of vectors that are (i) uniform in direction and (ii) have fixed norm values, become asymptotically Gaussian. Our analysis shows that InfoNCE drives representations toward these geometric conditions. This explains why contrastive representations exhibit approximately Gaussian distributions in practice, independently of whether the input distribution is Gaussian.

---

> ### Author Response · Authors · 2025-11-20
> **Response (3)**
>
> **Related to previous weakness and question**:
> Altogether, our empirical evidence spans (i) synthetic Laplace and Gaussian mixture data with a linear encoder, (ii) CIFAR-10 with a small MLP, and (iii) CIFAR-10 with ResNet-18 trained either contrastively (SimCLR-style) or with standard supervision. Across these settings we find that (a) our diagnostics are well defined and consistent, (b) Gaussianity and thin-shell concentration reliably arise under contrastive training even for highly non-Gaussian inputs, and (c) this behavior is not a generic consequence of LLN, SGD, or architecture, but is instead tied to the geometric bias of contrastive objectives.
>
> **Weakness:**
> "In particular, the raw unnormalized representations have received little theoretical attention": This statement seems ungrounded, indeed a lot of identifiability theory also studies the unnormalized case (see above).
>
> **Response:**
> The reviewer is correct that identifiability papers analyze encoder outputs in their raw (unnormalized) form. As we have explained above, our work is complementary this line of work. Nevertheless, we have clarified our statement in the main body (lines 91-92) to:
>
> > "In particular, there is little theoretical understanding of the distributional geometry of the raw, unnormalized representations, specifically their asymptotic laws."
>
> **Question:**
> What is the expectation in Eq. 8 over?
>
> **Response:**
> The expectation in Equation 8 is taken over the population distribution of samples and their augmented views, as in Equation 5. Concretely, it is over pairs generated by the data distribution composed with the augmentation channel. We have updated Equations 7 and 8 (lines 197, 230) in the paper to make this explicit in the notation, thank you for raising this question.
>
> **Question:**
> Consider a mixture of Gaussian distribution, or a vMF mixture where each component represents one class; can you comment if the marginal distribution expected through contrastive learning for this case would be Gaussian? Previous work seems to suggest that a mixture of Gaussians would be recovered, and this seems to conflict with the presented theory here; could you comment?
>
> **Response:**
> We thank the reviewer for this insightful question. We were not aware of prior identifiability results showing that, under certain assumptions, contrastive learning can recover latent mixture structure (e.g., class-conditional components of a Gaussian or vMF mixture) up to allowable transformations. However, as we understand, these works focus on the class-conditional or component-level geometry of the representation. That means they examine whether different mixture components remain distinguishable, not the marginal distribution of the embedding itself.
>
> Our analysis concerns a different object: the marginal law of the representation aggregated over all data points, irrespective of class or mixture membership. These two perspectives are not contradictory. A representation can be such that (i) mixture components remain identifiable and separable, while (ii) the overall embedding distribution converges toward a Gaussian shape.
> To illustrate this, we added a new experiment where the input data samples come from a synthetic Gaussian mixture. Even though the underlying mixture components are distinct in the input domain, the marginal distribution of the learned representations is approximately Gaussian according to our diagnostics (Table 1 and lines 445-455). This aligns with our theoretical predictions: the marginal distribution is shaped by the InfoNCE objective, not by the original input mixture law.
>
> In short, identifiability theory asks whether mixture components can be recovered; our work asks what the marginal distribution looks like after contrastive learning. These are complementary but conceptually separate questions, and the existence of identifiable mixture structure does not preclude the marginal distribution from converging to a Gaussian form. We have clarified this distinction in the revised Related Work section.
>
> **Final note:**
> We hope that our clarifications, revisions and the additional empirical results directly address the reviewer’s concerns regarding identifiability and the rigor of our experiments. We would be happy if the reviewer could positively reassess the paper in light of these explanations and enhancements. We appreciate the thoughtful feedback and believe the revised version is substantially strengthened as a result.

---

### Author Response · Authors · 2025-11-27

**Dear AC and Reviewers**,
We would like to briefly note that we have incorporated the reviewer feedback into the revised manuscript, including clarifications in the introduction and related work, additional experiments, and refined statements. The updates are marked in yellow and address each concern raised during the review.

We believe these revisions have improved the framing and motivation of the work, strengthened the empirical evidence, and clarified how our contributions relate to existing literature. We would be very grateful for any additional comments or discussion during the remaining rebuttal period. In particular, since these changes, including several experiments, were made directly in response to the reviews, it would be extremely helpful to know whether the main concerns now feel adequately addressed; we are happy to clarify any remaining points.

Thank you again for the time and care invested in reviewing and for the feedback that guided these revisions.

---

### Author Response · Authors · 2025-12-02

Dear AC,

We briefly summarize the reviews and the revisions performed during the rebuttal phase (highlighted in yellow). This is a theoretical paper, and importantly none of the reviewers raised concerns about the correctness of the theory, proofs, or technical claims, nor did any reviewer point to prior work that already establishes our results. All feedback focused instead on framing, connections to existing literature, and empirical evaluation. The comments were constructive, and the manuscript has been significantly strengthened. Two reviewers provided follow-up replies, both positive and explicitly noting the improvement, while the remaining two did not manage to reply before the discussion window closed unexpectedly. Below we outline the main points raised and how we addressed them:

**Relation to prior work and framing**:
Reviewers requested clearer relationships to identifiability work (j58r), uniform-sphere analyses (vUPT, j58r), and clustering/separability results (yqFV), along with softer claims (j58r) and clearer practical implications(yqFV). We expanded and reorganized the Related Work, explicitly contrasting our distribution perspective with identifiability and class-conditional analyses, and emphasized complementarity rather than contradiction. Reviewer j58r requested an additional experiment in a setting where no latent Gaussian model is recoverable; we conducted and added this in an additional response. We also softened relevant claims and added a discussion in the intorduction on practical uses of Gaussianity, referring to prior work and highlighting the advantages of Gaussian modeling over analyses that treat the representation distribution as arbitrary or unknown.

**Assumptions and realism**: Reviewers asked for clearer articulation of the strength, purpose, and practicality of the assumptions (vUPT, BFsh, yqFV), including how they relate and when they are expected to hold. In response, we reorganized the beginning of Sec. 4 to present this hierarchy explicitly, stating which assumptions describe plausible training behavior (plateau-based) and which are introduced solely to ensure the existence of a suitable solution in our analysis (regularized route). We clarified that the weaker assumption (regularized route) does not model the optimization trajectory but only requires that such a solution exists, and we added empirical evidence on alignment dynamics, norm concentration, and sensitivity to regularization.

**Metrics and diagnostics:** Reviewer j58r requested clearer definitions and justification of the statistical and geometric diagnostics we use to assess Gaussianity. In response, we now state the null hypotheses and acceptance thresholds for the normality tests, and explain why per-coordinate 1D tests, together with norm concentration, provide complementary finite-sample indicators of the asymptotic behavior predicted by our theory.

**Evaluations:** Reviewers (j58r, BFsh, vUPT) asked for a broader and more targeted empirical evaluation, including a SimCLR setting, controlled synthetic data, and checks that Gaussianity is not a trivial consequence of architecture, SGD, or high dimensionality. In response, we expanded the experiments to include: synthetic Gaussian mixtures; a discrete binary dataset where no continuous Gaussian latent model is recoverable; CIFAR-10 with ResNet-18 trained supervised or with SimCLR; augmentation-strength ablations; and checks on non-natural images (ImageNet-R). We also frame the DINO/CLIP experiments as evidence of generalization and as motivation for further work in self-supervised settings, rather than as direct single-encoder InfoNCE examples. Across these settings, we show that contrastive training consistently yields Gaussian-like, thin-shell embeddings, while supervised training on the same data and architecture does not, supporting the theoretical claims and clarifying the scope of their empirical validity.

**Normalized vs. unnormalized embeddings**: Reviewers (vUPT, yqFV) asked whether Gaussianity holds consistently in both the normalized and unnormalized representation spaces. In response, we clarified in Sec. 4 that both theoretical routes apply to both the normalized and the unnormalized representations. We also highlighted empirical evidence across all experiments, showing that both normalized and unnormalized embeddings exhibit strong Gaussianity, matching the predictions of the theory.

We would like to highlight that the reviewer feedback was focused and mostly directly actionable, and that we incorporated every raised point into the revised manuscript. The theoretical core of the paper was uncontested, and the revisions meaningfully improved the framing, connections to prior literature, and empirical scope. While some of the initial ratings were low, they were primarily driven by concerns that we believe are now explicitly addressed in the revised version. We therefore feel the paper is now significantly clearer and stronger.

---

### Public Comment · ~Ivan_Butakov1 · 2026-05-11
**Prior works on representations' distribution**

Dear Authors,

Thank you for this interesting work! I was glad to have the opportunity to attend your oral presentation in person.

I am writing this public comment to discuss a statement in the related work overview that I believe is not entirely accurate. At the end of the first paragraph of Section 2, you write:
> However, the probabilistic law induced by the InfoNCE objective itself remains theoretically unexplained.

While your asymptotic analysis of $k$-dimensional projections' distribution in the limit $d \to \infty$ is, to the best of my knowledge, indeed novel, the general assertion is too broad: several other works have already investigated the distributions induced by InfoNCE under various settings.

For instance, our ICLR 2025 paper, "Efficient Distribution Matching of Representations via Noise-Injected Deep InfoMax" [1], shows that with a small Gaussian noise injection, InfoNCE maximizes not only mutual information but also the entropy of the representations. This, in turn, Gaussianizes the distribution in the non‑asymptotic regime ($d=const$).

Additionally, a parallel line of work connects pre‑InfoNCE similarity measures (e.g., cosine similarity) to specific distributions induced by the loss, such as the von Mises-Fisher distribution. Some of these studies date back to 2021; for brevity, I refer here to a recent article that provides a decent overview [2].

Since the conference permits final revisions for the camera‑ready version after the event, I kindly ask you to update this part of the manuscript to reflect these contributions. Thank you!

[1] I. Butakov et al., "Efficient Distribution Matching of Representations via Noise-Injected Deep InfoMax". Proc. of ICLR 2025

[2] Fabian A Mikulasch, Friedemann Zenke, "Understanding Self-Supervised Learning via Latent Distribution Matching". arXiv:2605.03517, 2026

---

### Meta-Review · Area_Chair_ZLVq · 2026-01-07

**Summary:**

Key questions raised in initial reviews:
- Novelty and positioning of the theory: contrasting to identifiability theory (j58r) and known soap bubble effect (vUPT)
- Practicality of theoretical setup: strong assumptions over alignment & concentration (BFsh + vUPT), and whether the theory holds without the vanishing regularizer (vUPT + yqFV)
- Missing empirical validations: missing null hypothesis for the high dimensional distributions being tested (j58r), and known data-generating processes enabling control over Gaussianity (j58r), and importantly missing experiments on SimCLR (BFsh)
- Other technical issues: DINO isn't quite contrastive but still shows Gaussianity (BFsh + vUPT), the dual-encoder (vision + text) setup for CLIP, and potential impact for downstream performance (vUPT + yqFV)

**Reviewer Concerns:**

Authors adequately addressed most key concerns, specifically:
- Clarified relationship to identifiability, supported by a binary dataset experiment showing Gaussianity emerges from loss geometry rather input geoemtry
- Clarified relationship to soap bubble, explaining why InfoNCE satisfies Maxwell-Poincaré CLT, beyond merely arguing this property holds
- Included SimCLR and compared supervised and contrastive models, specifically acknowledging that DINO doesn't quite fit the theory, together demonstrating the theory holds more broadly
- Other clarifications: hierarchy of assumptions and sensitivity analysis of the proposed regularizer, also softening the tone "first-principled" (following discussion with reviewer j58r)

Overall, InfoNCE is indeed a fundamental algorithm for unsupervised learning with far reaching impact through the widely adopted CLIP.  The theoretical insights, and supporting empirical evidence, sheds light on the learned embeddings and offers new tools for a variety of tasks facilitated by the proxy Gaussian model established in this work.  Clarifying the setup and positioning by contrasting to related approaches, as can be seen in the discussion with reviewers, further highlights the value of this piece of work.  This should make for a decent paper at ICLR this year.

**Reviewer Scores:**

Initial ratings came as 8/4/2/2.  While we cannot be sure how reviewers may have updated their scores, I'd expect a final score above 6.

---

### Decision · Program_Chairs · 2026-01-26

Accept (Oral)